# Defining the proteolytic landscape during enterovirus infection

Mohsan Saeed[1,2,3]*, Sebastian Kapell[1,2,3,4], Nicholas T. Hertz[5,6], Xianfang Wu[1], Kierstin Bell[1], Alison W. Ashbrook[1], Milica Tesic Mark[7], Henry A. Zebroski[7], Maxwell L. Neal[8], Malin Flodström-Tullberg[4], Margaret R. MacDonald[1], John D. Aitchison[8,9,10], Henrik Molina[7], Charles M. Rice[1]*

1 Laboratory of Virology and Infectious Disease, The Rockefeller University, New York, NY, United States of America, 2 Department of Biochemistry, Boston University School of Medicine, Boston, MA, United States of America, 3 National Emerging Infectious Diseases Laboratories, Boston University, Boston, MA, United States of America, 4 The Center for Infectious Medicine, Department of Medicine HS, Karolinska Institute, Karolinska University Hospital, Stockholm, Sweden, 5 Laboratory of Brain Development and Repair, The Rockefeller University, New York, NY, United States of America, 6 Department of Biology, Stanford University, Stanford, CA, United States of America, 7 Proteomics Resource Center, The Rockefeller University, New York, NY, United States of America, 8 Center for Global Infectious Disease Research, Seattle Children's Research Institute, Seattle, WA, United States of America, 9 Departments of Pediatrics and Biochemistry, University of Washington, Seattle, WA, United States of America, 10 Institute for Systems Biology, Seattle, WA, United States of America

* msaeed1@bu.edu (MS); ricec@mail.rockefeller.edu (CMR)

**Data Availability Statement:** All relevant data are within the manuscript and its Supporting Information files.

**Funding:** This work was supported in part by the National Institutes of Health, NIAID grants R01

## Abstract

Viruses cleave cellular proteins to remodel the host proteome. The study of these cleavages has revealed mechanisms of immune evasion, resource exploitation, and pathogenesis. However, the full extent of virus-induced proteolysis in infected cells is unknown, mainly because until recently the technology for a global view of proteolysis within cells was lacking. Here, we report the first comprehensive catalog of proteins cleaved upon enterovirus infection and identify the sites within proteins where the cleavages occur. We employed multiple strategies to confirm protein cleavages and assigned them to one of the two enteroviral proteases. Detailed characterization of one substrate, LSM14A, a *p* body protein with a role in antiviral immunity, showed that cleavage of this protein disrupts its antiviral function. This study yields a new depth of information about the host interface with a group of viruses that are both important biological tools and significant agents of disease.

## Author summary

Enteroviruses are associated with a variety of human diseases, including gastroenteritis, the common cold, hand-foot-and-mouth disease, acute hemorrhagic conjunctivitis, and skin rash. In some cases, the infection can lead to myocarditis, encephalitis, progressive muscle weakness, and paralysis. Exactly how enteroviruses invade human tissues, defeat the host immune system, and alter normal cell biology is unknown. Understanding these cellular and molecular mechanisms will blaze the trail for the development of novel vaccine and therapeutic strategies. Here, we have applied a global N-terminomics approach

AI091707 (to C.M.R.), NCDIR grant
2P41GM109824-06 and R01 2R01GM112108-05
(to J.D.A.), and an Institut Mérieux grant (to M.S.
and C.M.R.). M.F-T was supported by Novo Nordic
Foundation, Swedish Medical Research Council,
and Karolinska Institutet's Strategic Research
Program in Diabetes. The funders had no role in
study design, data collection and analysis, decision
to publish, or preparation of the manuscript.

**Competing interests:** The authors have declared
that no competing interests exist.

to investigate how various enteroviruses recruit their proteases to remodel an infected cell, disarm host immunity, and create a favorable environment for their replication. This effort identified several new protease substrates, which we then confirmed by other experimental approaches. To our knowledge, this is the first systematic analysis of host proteins targeted for cleavage during enterovirus infection. The data generated in this study will serve as a valuable resource for the research community in the quest to uncover the molecular details of enterovirus cell biology and disease pathogenesis.

## Introduction

Viruses hijack or subvert host cell machinery to create a favorable environment for their replication. One way they do this is by cleaving host proteins. A number of viruses encode one or more proteases that cleave cellular proteins during infection in a spatially and temporally regulated manner, while other viruses recruit host proteases to promote their replication. Over the years, numerous proteins involved in diverse cellular processes including gene expression, autophagy, apoptosis, vesicular transport, and antiviral immunity have been identified as proteolytic targets upon virus infection [1–4]. The study of these cleavages has helped unravel the complex interplay between viruses and their hosts, as well as advanced our understanding of cell biology. However, technical limitations have meant that targets of viral proteases are generally identified one at a time, so the full extent of proteolysis induced upon infection remains unknown. In this study, we report the first comprehensive view of host proteins cleaved upon enterovirus infection.

Enteroviruses are one of the leading causes of human disease worldwide. Classified in the *Picornaviridae* family, the enterovirus genus comprises over 300 genotypes grouped into 10 enterovirus (EV) species (EV-A to EV-J) and three rhinovirus (RV) species (RV-A to RV-C) (reviewed in [5]). The enteroviruses cause a range of diseases, including the common cold (human rhinoviruses [HRV]), hand-foot-and-mouth disease (Coxsackievirus, enterovirus A71 [EV71], and others), acute hemorrhagic conjunctivitis (enterovirus D70 [EV70]), poliomyelitis (poliovirus), and recent outbreaks of acute flaccid paralysis (possibly associated with enterovirus D68 and others). In some cases, infection can lead to severe neurological disease, myocarditis, and encephalitis, and these viruses have been implicated in epidemics and deaths worldwide. Enteroviruses, such as poliovirus, have also been used for decades as tools to probe virus-host interactions.

Enteroviruses possess a positive-stranded RNA genome of around 7,500 nucleotides that is composed of a long open reading frame (ORF) flanked by untranslated regions (UTR). The 5' UTR contains an internal ribosomal entry site (IRES) that drives cap-independent translation of the ORF to generate a large precursor polyprotein [6, 7]. Embedded in this polyprotein are two proteases, 2A (2A$^{pro}$) and 3C (3C$^{pro}$), that cleave the polyprotein co- and post-translationally to liberate 11 mature proteins [8, 9]. In the context of viral polyprotein processing, 2A$^{pro}$ is only responsible for its own cleavage from the upstream structural protein [10], and all but one of the remaining cleavages are catalyzed by 3C$^{pro}$ [8]. The final cleavage occurs by an autocatalytic mechanism [11].

Enteroviral proteases cleave host proteins as well. This was first described during the 1980s when poliovirus 2A$^{pro}$ was shown to cleave eIF4G, an essential translation initiation factor required for cap-dependent translation of host mRNAs [12]. As a result of eIF4G cleavage, host translation shuts off, while cap-independent viral RNA translation remains intact. It is believed that by blocking host translation, poliovirus and other enteroviruses suppress the expression of antiviral genes and thereby create a favorable environment for virus replication.

Since the discovery of eIF4G cleavage, several more cellular proteins have been identified as enteroviral protease substrates. However, most of these proteins were discovered by either candidate approaches, two-dimensional (2D) gel electrophoresis, or *in silico* predictions [13, 14]. These methods, while useful, are limited in their ability to provide a global view of proteolysis in a cell.

With recent advances in proteomics, several methods are now available that allow unbiased labeling of newly generated protein N-termini (neo N-termini), such as those resulting from proteolytic cleavage [15–17]. These methods leverage the fact that ~85% of the nascent protein N-termini in mammalian cells are post-translationally modified, mainly by acetylation, and are therefore unavailable for labeling [18]. One labeling method utilizes a bioengineered protein ligase, subtiligase, to append a biotinylated peptide to the neo N-termini produced inside the cells [16]. The biotinylated proteins are then captured on streptavidin beads, subjected to on-bead trypsinization, released from the beads through tobacco etch virus (TEV) protease cleavage, and identified by highly sensitive mass spectrometry (LC-MS/MS). The biotinylated peptide is designed such that after being removed by the TEV protease, it leaves a non-natural amino acid mass tag (aminobutyric acid, or Abu) attached to the neo N-terminal end, enabling high-confidence identification of the labeled peptides over nonspecifically bound background [19]. This method has been extensively used to study caspase-mediated proteolysis during apoptosis [16, 19, 20]. Another N-terminomics approach was recently used to investigate cellular proteins cleaved by enteroviral 3C$^{pro}$ in cell lysates [21], but this technique has not been attempted in the context of infection.

In this report, we adapted the subtiligase labeling approach for global identification of proteins cleaved upon virus infection. By applying this method to five viruses from diverse enteroviral species, we identified numerous known and novel cellular targets that we then validated with orthogonal approaches. Among the newly identified cleavage substrates was LSM14A, a *p* body-resident protein previously implicated in defense against some RNA and DNA viruses [22]. We show that the enteroviral 2A$^{pro}$-mediated cleavage of this protein prevents it from boosting antiviral defenses.

## Results

### Hundreds of host protein cleavage events detected after Coxsackievirus B3 infection

To map the landscape of enterovirus-induced proteolysis, we adapted a subtiligase method for labeling newly-cleaved N-termini in infected cells [20]. We chose to use Coxsackievirus B3 (CVB3) for initial analysis, as its known proteolytic targets could be used for validation. As a first step, we optimized the protocol to reduce the number of cells required and to allow simultaneous handling of multiple samples, which were prerequisites for time course analysis of virus infection with multiple replicates. Sonication of cells in buffer containing 1% SDS, followed by depleting the lysates of detergent prior to the labeling reaction, ensured optimal lysis (as measured by histone H3 solubilization, S1A Fig) and robust subtiligase-mediated biotinylation (S1B Fig). Optimization of post-lysis steps showed that 3 mg cellular protein per labeling reaction was sufficient for identification of the known enteroviral targets, as opposed to 30 mg per reaction reported previously for detection of caspase-mediated cleavages [20]. Similarly, we optimized the wash conditions to achieve greater signal over background. Next, we determined the timing of HeLa cell infection using CVB3 encoding an enhanced green fluorescent protein (eGFP) reporter gene (eGFP/CVB3). GFP-positive cells were first detected at around 3 hours post-infection (h.p.i.), and the infection reached its peak within the next hour (S2A Fig). The cytopathic effects (CPE) of infection were obvious by 6 h.p.i.

The optimized subtiligase method was used to label free N-termini in uninfected and CVB3-infected HeLa cells at various times post-infection (Fig 1A). Newly-cleaved peptides

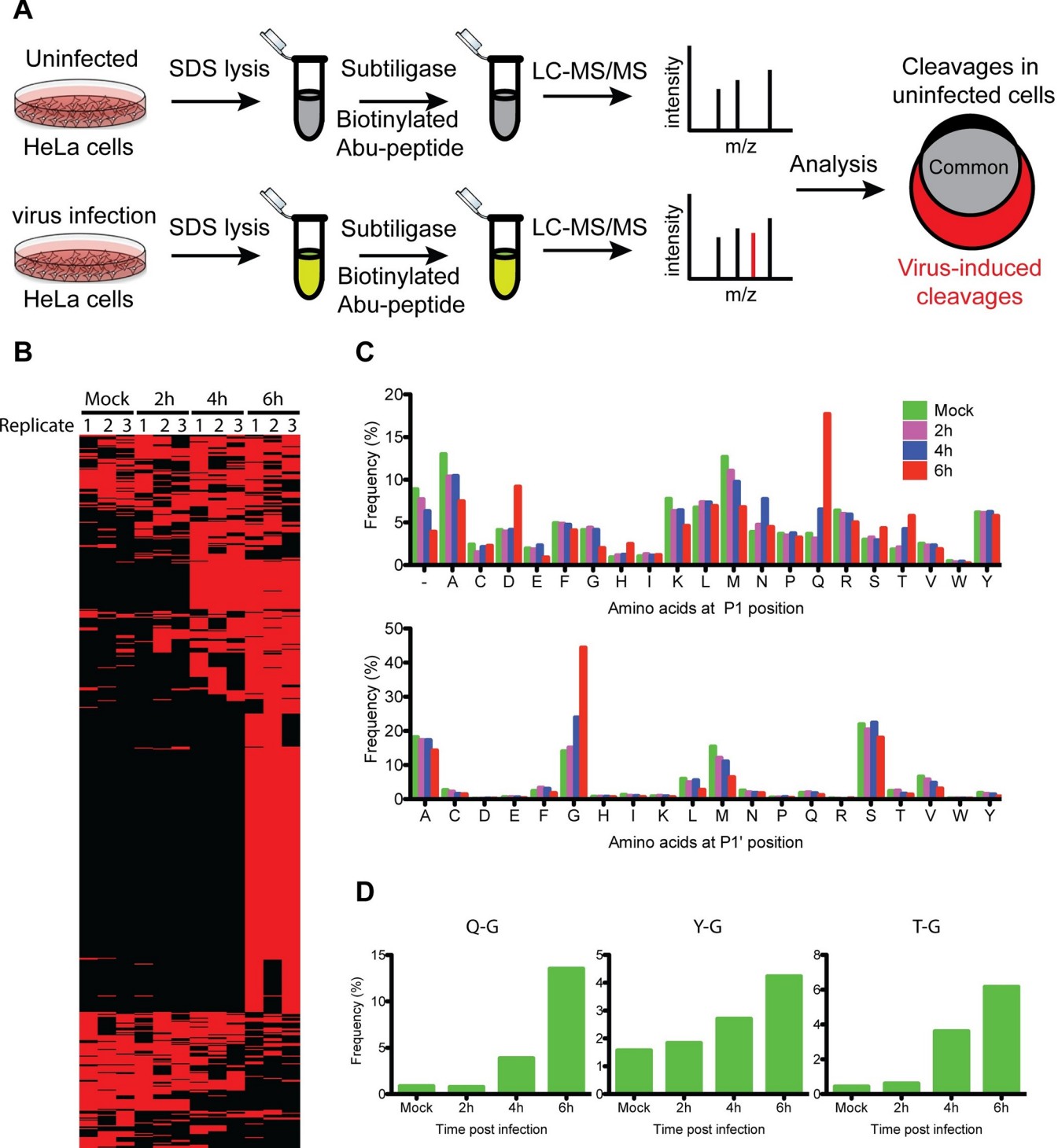

**Fig 1. Analysis of protein cleavages in HeLa cells upon eGFP/CVB3 infection.** (A) Subtiligase labeling workflow. HeLa cells were either infected with a virus or left uninfected, lysed in SDS buffer, and the lysates containing 3 mg total protein were subjected to subtiligase enrichment of protein N-termini. LC-MS/MS identification and bioinformatics analysis of the enriched N-termini yielded a list of virus-induced protein cleavages. (B) HeLa cells infected with eGFP/CVB3 were harvested in triplicate at the indicated times post-infection followed by subtiligase labeling. Each row indicates a unique Abu-labeled peptide, and the red and black colors are used to show the presence and absence of the peptide, respectively. The peptides are presented in no particular order. (C) Frequencies of P1 and P1′ amino acids from unique N-termini identified at the indicated times post-infection. The N-termini present in at least two of the three replicates were included in this analysis. (D) Frequencies of protein cleavages derived from the indicated P1 and P1′ amino acid pairs at various times post-infection. Only the cleavages seen in at least two of the three replicates were counted.

in each lysate were analyzed by LC-MS/MS, with triplicate samples for each time point. From this experiment, we identified many free N-termini in both infected and uninfected cells. Interestingly, the magnitude of labeling increased as the infection progressed, suggesting an increase in the number of neo N-termini (Fig 1B and S2B Fig). Indeed, considering only peptides that appeared in at least two of the three replicates for LC-MS/MS, the number of label-accessible peptides in the infected cells increased from 868 (at 2 h.p.i.) to 995 (at 4 h.p.i) to 1458 (at 6 h.p.i) (Table 1). Out of the 995 peptides identified at 4 h.p.i., 455 were found only in the infected cells, and the number of infection-specific peptides increased to 978 at 6 h.p.i.

Viral and cellular proteases exhibit cleavage site preferences, allowing the terminal residues of the cleavage products to provide clues about the identity of the protease. Enteroviral 2A$^{pro}$ and 3C$^{pro}$ are chymotrypsin-like proteases with a cysteine nucleophile and cleave the viral polyprotein at consensus motifs mainly defined by the P1 and P1′ residues. 3C$^{pro}$ exhibits strict site specificity for glutamine (Q) at the P1 position and glycine (G) > alanine (A) > serine (S) at the P1′ position [8, 13]. Interestingly, 21% of N-termini identified in CVB3-infected cells at 6 h.p.i. were derived from P1 glutamine cleavage, in contrast to 4% in uninfected cells (Fig 1C: top panel and Table 1). In addition, we observed 10- and 30-fold more Q-G cleavages at 4 and 6 h.p.i., respectively, compared to uninfected cells (Fig 1D: left panel, and Table 1). The number of Q-A and Q-S cleavages also increased upon infection (Table 1). In contrast to 3C$^{pro}$, 2A$^{pro}$ can tolerate multiple P1 residues, although it prefers threonine (T), tyrosine (Y), and phenylalanine (F). It however has a strict requirement for G at the P1′ position. In line with this, P1′ glycine cleavages increased by 3-fold upon virus infection (Fig 1C: bottom panel, and Table 1). As the infection progressed, more cleavages were observed at the 2A$^{pro}$ motifs (Fig 1D: middle and right panels, and S3 Fig). The greatest increase was seen for T-G cleavages, which rose from 0.2% at 2 h.p.i. to 4.5% at 6 h.p.i. (Fig 1D: right panel). Although some of these cleavages might be catalyzed by host proteases, the fact that they were seen at generally much lower levels in the uninfected cells supports the hypothesis that they were mediated by the viral proteases.

**Table 1. Summary of CVB3 induced proteolysis in HeLa cells.** The total numbers of peptides and the corresponding proteins in the mock and virus-infected cells are tabulated. For several proteins, multiple peptides were identified, suggesting that those proteins were cleaved at more than one site. Also, the numbers of cleavages at the indicated motifs are shown.

| Cleavages (P1\|P1′ residues) | Mock Peptides (proteins) | 2h Peptides (proteins) | 4h Peptides (proteins) | 6h Peptides (proteins) |
|---|---|---|---|---|
| Total | 876 (702) | 868 (698) | 995 (776) | 1458 (1087) |
| Q\|X** | 32 (31) | 27 (27) | 65 (61) | 258 (239) |
| X\|G | 123 (107) | 131 (116) | 238 (203) | 647 (518) |
| D\|X | 36 (32) | 34 (28) | 41 (36) | 134 (124) |
| Q\|G | 3 (3) | 3 (3) | 29 (26) | 148 (137) |
| Q\|A | 6 (6) | 6 (6) | 8 (8) | 36 (36) |
| Q\|S | 14 (14) | 11 (11) | 17 (17) | 58 (58) |
| T\|G | 2 (2) | 3 (3) | 26 (26) | 66 (62) |
| Y\|G | 11 (10) | 10 (10) | 20 (18) | 41 (39) |
| F\|G | 6 (6) | 8 (8) | 15 (15) | 24 (23) |
| V\|G | 2 (2) | 3 (3) | 3 (3) | 9 (9) |
| A\|G | 11 (11) | 10 (10) | 10 (10) | 30 (30) |

** any amino acid

At later stages of infection, enteroviruses are known to induce apoptosis. The effector proteases in the apoptotic pathway, caspases, cleave their targets between aspartate (D) and small amino acids (G > S > A) [23, 24]. At 6 h.p.i., when the cells exhibited substantial CPE, 134 of the neo N-termini were derived from P1 aspartate cleavage, in contrast to 41 at 4 h.p.i (Table 1). This suggests that caspases are also impacting the proteolytic landscape of CVB3-infected cells.

Overall, these results indicate that hundreds of cleavages take place in CVB3-infected cells, and that many of the resulting peptides show the signatures of 2A$^{pro}$, 3C$^{pro}$, and caspase cleavage. Furthermore, these results suggest that the modified subtiligase labeling conditions are well suited for identifying protein cleavages in virus-infected cells.

## Multiple novel targets of CVB3 proteases identified and validated

As an initial validation of the subtiligase labeling approach, we inspected the dataset of all newly-cleaved proteins for the presence of known CVB3 targets. CVB3 has been reported to cleave a number of host proteins, including dystrophin, eIF4G, HNRPD, HNRPM, MAVS, NUP98, PABP, RIP3, and TRIF [1]. Peptides from the majority of these proteins were present in our dataset, with the exceptions of RIP3, which is not expressed in HeLa cells [25], and dystrophin. Most importantly, the peptides for eIF4G, HNRPD, HNRPM, MAVS, and PABP corresponded to the known cleavage sites in these proteins, although additional peptides were also detected. These findings supported the validity of our labeling approach.

Next, we surveyed our dataset for the presence of novel cleavage targets. To ensure high-confidence identification of targets, we included only those proteins that were detected in at least two of the three infected, but none of the uninfected, replicates. Similarly, to exclude non-specific cleavages, we included proteins that were cleaved early during infection before the cells reached cytotoxicity. This yielded a list of 173 proteins (S1 File), of which around 81 were cleaved at motifs known to be targeted by CVB3 proteases. These proteins are involved in numerous cellular pathways, including transcription, RNA editing, splicing, transport, and turnover, cytoskeleton maintenance, cell division, DNA repair, endocytosis and secretion, and innate immunity. We chose 20 of these 81 proteins, along with 14 late-stage targets (ARFP1, CAPR1, CHERP, DBNL, F120A, HTSF1, MAPK3, MATR3, NUFP2, PANX1, RBP56, SHRM1, SP130, and STAT3) and six known targets (EIF4G, HNRPD, HNRPM, MAVS, NUP98, and USO1) identified in our dataset, for validation studies (Fig 2B).

To validate cleavage of the candidate target proteins in the presence of virus, we infected HeLa cells with eGFP/CVB3 and monitored the proteins over time by western blot (Fig 2A and S4 Fig). Remarkably, all proteins tested were cleaved, with around 75% of them yielding the banding pattern consistent with the cleavage site(s) identified by degradomics. For some proteins, the full-length molecules decreased in abundance without yielding a detectable cleavage product. This is likely due to alteration of the epitope recognized by the antibodies used or progressive degradation of the cleavage products. To test if the epitope alteration was responsible for the lack of detectable cleavage product, we generated doubly-tagged versions of some proteins fused at their N-terminal end with a V5 tag and at their C-terminal end with an HA tag. Should the protein be cleaved, western blot with anti-V5 and anti-HA antibodies would identify N-terminal and C-terminal cleavage products, respectively. Of the five proteins tested, four had detectable cleavage products (S5 Fig).

Importantly, the western blot results confirmed the timing of proteolytic cleavage as observed in the proteomics dataset. EIF4G and NUP98 were cleaved within 2h of infection, while other proteins were mostly cleaved at 4 h.p.i. This was the time point at which GFP became detectable, suggesting an overlap between virus replication and protein cleavage (Fig 2B). For 12 proteins, cleavage was detected only at 5 h.p.i. Among these, seven were consistent

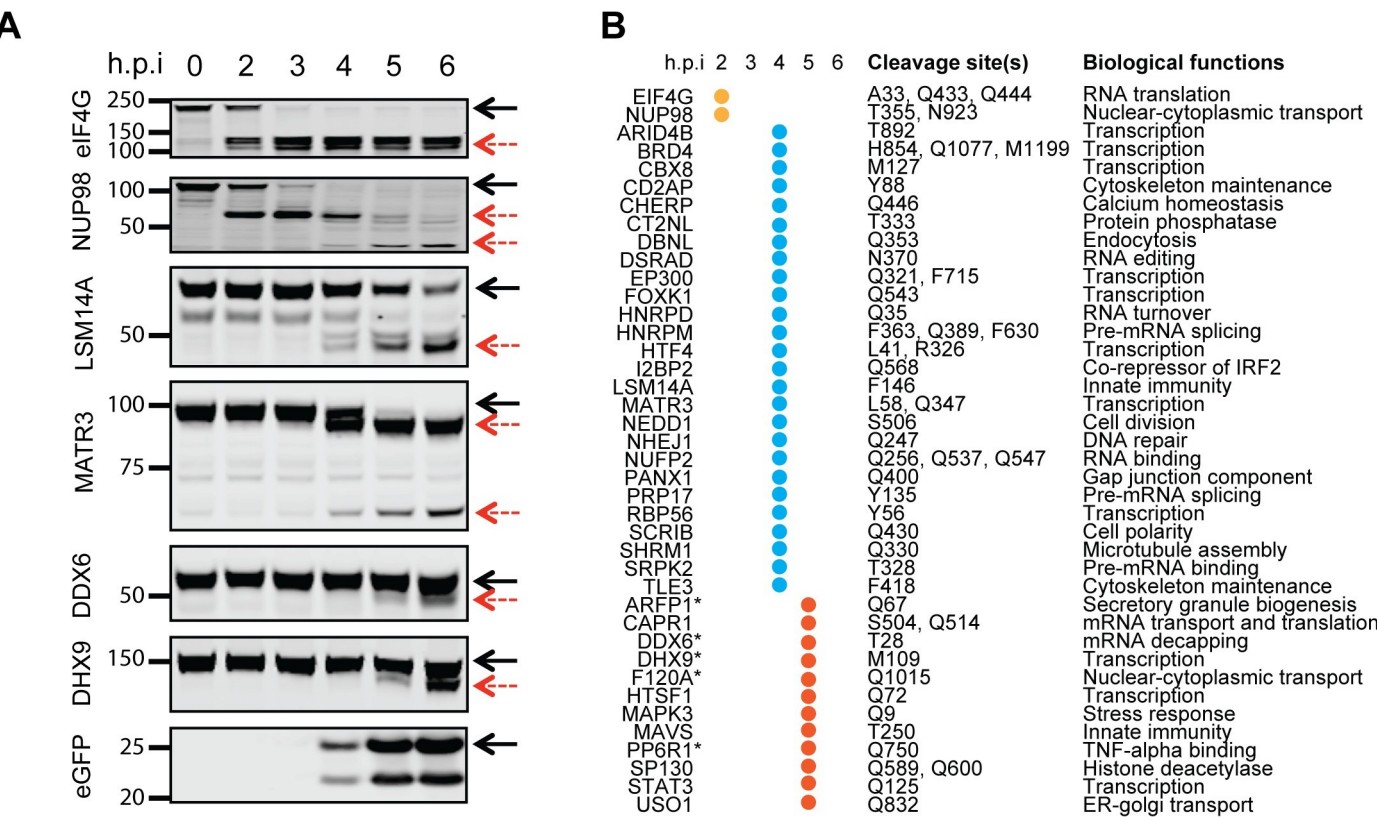

**Fig 2. Validation of protein cleavages identified by subtiligase labeling.** (A) HeLa cells were infected with eGFP/CVB3 and lysed at 0, 2, 3, 4, 5, and 6 h.p.i. followed by western blot analysis of the indicated proteins. An equal amount of total protein, as quantified by the BCA assay, was loaded for each time point. GFP expression was used to monitor the progression of infection (bottom panel). The black solid arrows indicate the full-length protein, while the cleavage products are shown with red dotted arrows. (B) The post-infection time point when the cleavage first became detectable is shown (the western blot images are shown in S4 Fig). Some proteins (CAPR1, HTSF1, MAPK3, MAVS, SP130, STAT3, and USO1) that were found to be cleaved only at 6 h.p.i. by proteomic analysis were included as controls. The proteins with asterisks were found to be cleaved at 4 h.p.i. by proteomics analysis but only at 6 h.p.i. by western blot. The position and identity of P1 residues identified at either 4 or 6 h.p.i. and the biological functions of the cleaved proteins are also shown.

with the proteomics results, while the remaining five were found to be cleaved earlier in the proteomics experiment, suggesting that subtiligase enrichment can be more sensitive than western blot. Overall, these results suggest that N-terminal subtiligase labeling is a robust method to identify infection-associated cleavage targets and that labeling correlates well with the timing of cleavage events.

## Purified recombinant CVB3 proteases cleave identified target proteins *in vitro*

To test if CVB3 proteases were responsible for the cleavages identified, we performed an *in vitro* cleavage assay. For this, we incubated HeLa cell lysates with purified recombinant CVB3 2A$^{pro}$ and 3C$^{pro}$ proteases, or their catalytically inactive versions, and monitored the cleavage of 38 of the identified substrates by western blot (Fig 3 and S6 Fig). It should be noted that although this method has previously been used to test protein cleavage [21, 26], the results can be heavily influenced by the quality of the protein preparations, buffer composition, the protease-to-protein ratio, and the reaction conditions. Therefore, we first optimized all of these parameters. Preparation of cell lysates in a mild detergent (0.1% Triton-X) and incubation of lysates with viral proteases at a protease to protein ratio of 1:200 for 2A$^{pro}$ and 1:4 for 3C$^{pro}$ yielded optimal results.

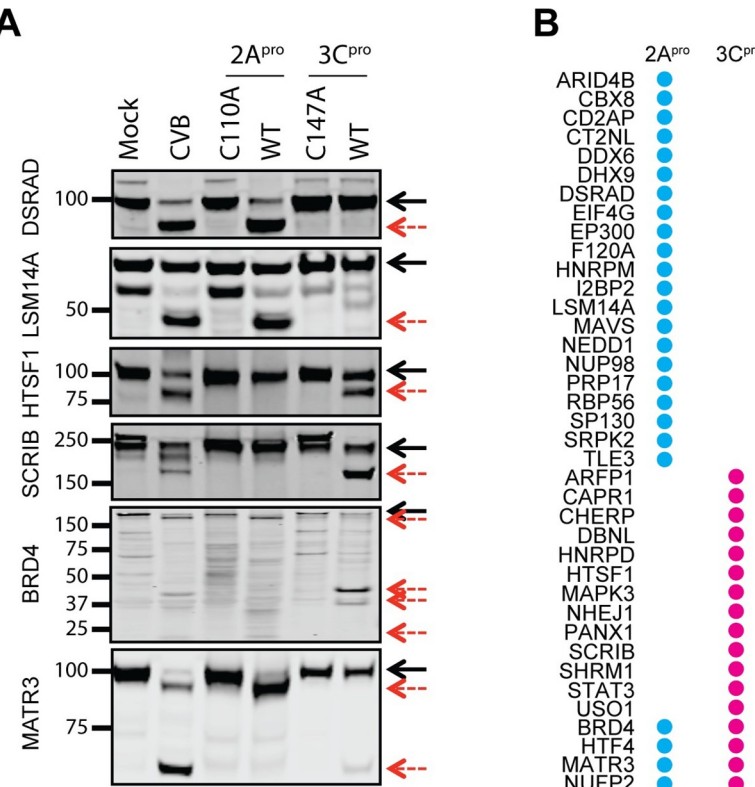

**Fig 3. *In vitro* cleavage assay of the identified proteins.** (A) HeLa cell lysates (200 μg protein) were incubated with CVB3 2A^pro or its catalytically inactive mutant C110A (1 μg), or CVB3 3C^pro or its catalytically inactive mutant C147A (100 μg) at 37˚C for 3h and analyzed by western blot. Lysates from the uninfected and eGFP/CVB3-infected HeLa cells were included as positive controls. The full-length protein is indicated with black solid arrows, while the cleavage products are indicated with red dotted arrows. (B) The proteins are grouped into three categories based on if the cleavage was mediated by 2A^pro, 3C^pro or both (the western blot images are shown in S6 Fig).

The *in vitro* assay showed that the presence of CVB3 proteases resulted in cleavage of all 38 proteins monitored. The results of the *in vitro* assay largely agreed with the protease suscepti- bility predicted from the cleavage sites. In general, sites with a P1 glutamine were cleaved by 3C^pro, while the rest were catalyzed by 2A^pro. While most of the proteins underwent cleavage only once, some were targeted multiple times either by the same or different proteases. For example, HNRPM appeared to be repeatedly targeted by 2A^pro (S6 Fig, 4^th lane, middle panel). In contrast, MATR3, which yielded two C-terminal cleavage products in the infected cells, was independently targeted by both 2A^pro and 3C^pro at two different sites (Fig 3). Consistent with the infection results, some proteins (ARFP1, F120A, HTF4, NHEJ1, NUFP2, PANX1, and SP130) decreased in abundance but did not produce detectable cleavage products. In all, these results suggest that the viral proteases were responsible for most of the cleavages identified at early time points in CVB3-infected cells.

## Identification of proteins targeted for cleavage by multiple enteroviruses

We were interested in comparing the proteolytic landscape of CVB3 infection to that of other enteroviruses, and to determine if common targets of proteolysis exist. To do this, we used a panel of enteroviruses associated with various human disease states: human rhinovirus A16 (HRV), poliovirus 1 (PV), enterovirus D70 (EV70), and enterovirus A71 (EV71). First, we tested the growth kinetics of these viruses in HeLa cells by monitoring the accumulation of

viral proteins (S7 Fig). PV exhibited the same characteristics as CVB3, with most of the cells expressing the viral antigens at 4 h.p.i. and exhibiting cytotoxicity at 6 h.p.i. EV70 and EV71 had slower growth kinetics, reaching peak infection at 6 h.p.i. HRV was the slowest of all with the highest number of infected cells seen at 9 h.p.i. Based on this, we selected 3, 4, and 6 h.p.i. time points for PV, 4, 6, and 8 h.p.i. for EV70 and EV71, and 6, 9, and 12 h.p.i. for HRV. For each virus, uninfected HeLa cells served as a control.

Proteomic analysis of subtiligase-labeled neo N-termini from cells infected with these viruses revealed 218 (PV), 278 (HRV), 274 (EV70), and 206 (EV71) candidate host targets of cleavage. This was comparable to the 211 host proteins identified for CVB3 (Fig 4A and S2 File). As expected from the structural and functional similarities of enteroviral proteases [1], many of the identified target proteins overlapped between one or more viruses; 46 proteins were detected as cleaved by all viruses tested. Some cleavages appeared specific to a single virus or a subset of viruses; however, unless confirmed by orthogonal approaches, it is difficult to

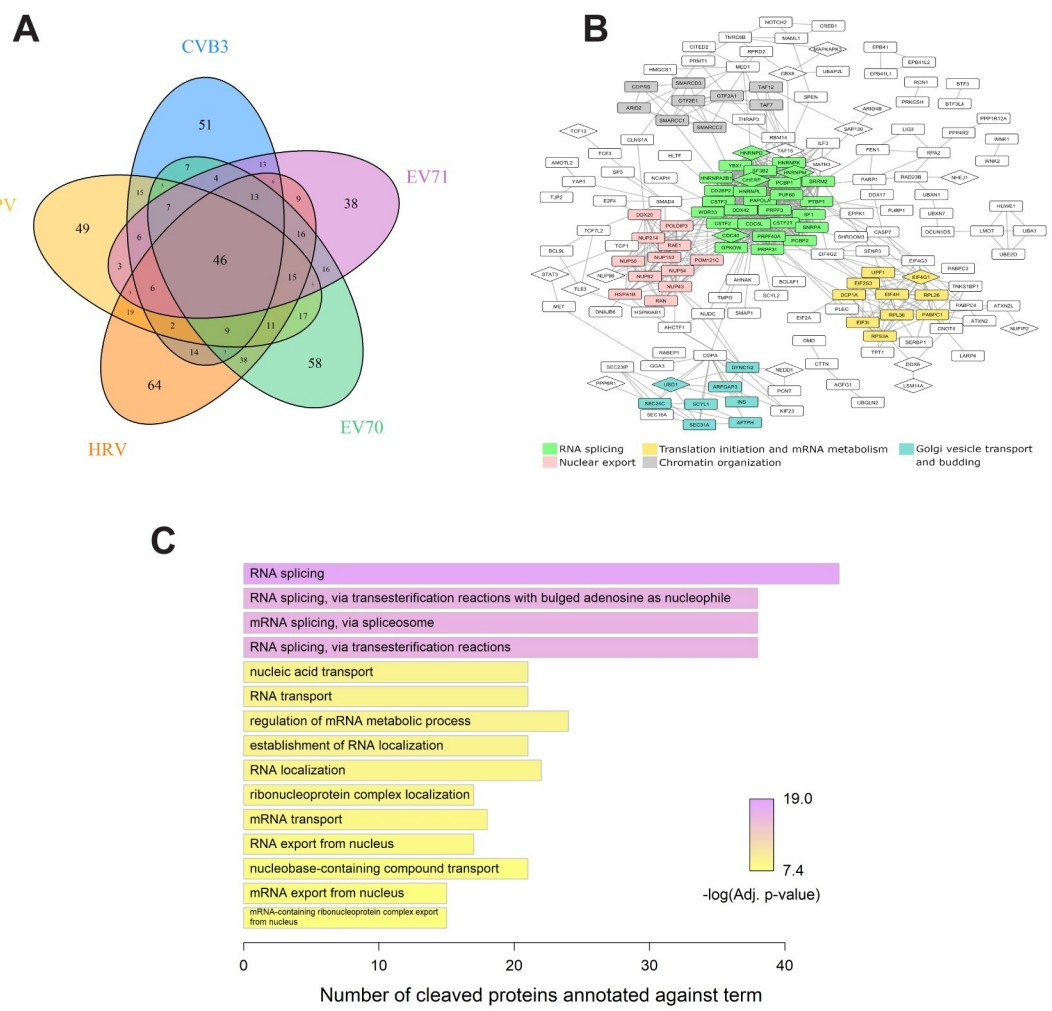

**Fig 4. Analysis of proteins broadly targeted by *enteroviruses*.** (A) Venn diagram showing the numbers of cleaved proteins common or unique to different viruses. (B) STRING interaction network of proteins targeted by at least two of the five enteroviruses as rendered in Cytoscape (disconnected nodes are not shown). Coloring indicates groups of five or more proteins identified as highly interconnected by the MCODE clustering algorithm. Most significantly enriched biological processes are summarized below the network. Cleaved proteins validated by western blot are shown in diamond shaped nodes. (C) Top 15 significantly enriched biological processes targeted by at least two of the five viruses.

conclude with confidence that a particular cleavage is virus-specific. This is because subtiligase labeling, like other proteomics-based approaches, may not identify all protein cleavages across samples, and peptides could also be excluded by the strict data analysis pipeline. Protein clustering and gene ontology (GO) enrichment analysis on proteins identified with high confidence for at least two of the five viruses (Fig 4B and 4C) showed enrichment of proteins from gene expression pathways, such as RNA splicing, mRNA export from the nucleus, translation initiation, and RNA metabolism. This aligned with previous observations that enteroviruses strongly inhibit gene expression (reviewed in [27]).

To identify common substrates of enterovirus proteases, and to narrow the candidates to a manageable number of samples, we focused on the 40 proteins that we had previously validated for CVB3. Most of these proteins were detected as cleaved in all proteomics datasets obtained, indicating that they were targeted by all viruses tested (S8 Fig). Some proteins (I2BP2, DSRAD, and MAVS) appeared to be cleaved by a subset of viruses, with two (DDX6 and DHX9) detected in only CVB3- and EV71-infected cells. Importantly, for most proteins, the same peptides were detected in the datasets of all viruses tested, indicating specific cleavage instead of non-specific degradation.

To validate the proteomics results for the panel of enteroviruses, we infected HeLa cells with each virus and tested cleavage of the 40 candidate targets by western blot at two different times post-infection (Fig 5 and S9 Fig). To determine whether diverse viral proteases also cleaved these substrates, we infected cells with two non-enteroviruses: Venezuelan equine encephalitis virus (VEEV), a positive-strand RNA virus from the *Togaviridae* family, and vesicular stomatitis virus (VSV), a negative-strand RNA virus from the *Rhabdoviridae* family. The western blot results largely confirmed the cleavage patterns revealed by the subtiligase labeling experiments. For example, consistent with the proteomics results, DDX6 and DHX9 were exclusively cleaved by CVB3 and EV71, although a faint band indicating cleavage at a different site was noted for DHX9 in PV-infected cells (Fig 5). In contrast to the proteomic results, western blot showed that MAVS was targeted by all viruses tested and, interestingly, seemed to be cleaved at different sites by different enteroviruses. This was also true for F120A, where more prominent cleavage bands were seen for HRV compared to CVB3 and PV, despite the fact that all of these viruses caused a similar decrease in the full-length protein (S9 Fig). This may be due to differential stability of the cleavage products generated by different viruses. For the most part, VEEV and VSV infection did not lead to cleavage of the cellular proteins tested, with the exception being incomplete cleavage of CT2NL by VSV and some degradation of MAVS by both viruses. Overall, these results again indicate that subtiligase labeling is a powerful technique to identify protein cleavage events associated with virus infection, and that the cleavages identified are largely specific to enterovirus infection as opposed to pan-viral.

Since our analysis had so far been limited to HeLa cells, we next investigated proteolysis of the identified targets by eGFP/CVB3 and PV in various cell types (S10 Fig). The panel of cell lines included Caco-2, an intestinal epithelial line representing the primary portal of CVB3 and PV entry into human body, RD, a skeletal muscle line representing one of the major sites of PV replication *in vivo*, and SK-N-SH, a neuronal line likely relevant for the neurotropic potential of enteroviruses. We also infected embryonic stem cell-derived neural progenitor cells (NPC) to test protein cleavage in a more physiologically relevant setting. Since CVB3 did not infect RD cells and only minimally infected SK-N-SH cells, we excluded these cells from CVB3 analysis. Notably, the selected proteins were found to be cleaved in all cell types tested, and the banding patterns mostly resembled those seen in HeLa cells. Taken together, the results so far suggest that enteroviruses mostly cleave an overlapping set of proteins, and that these cleavages are consistent across a variety of cell types.

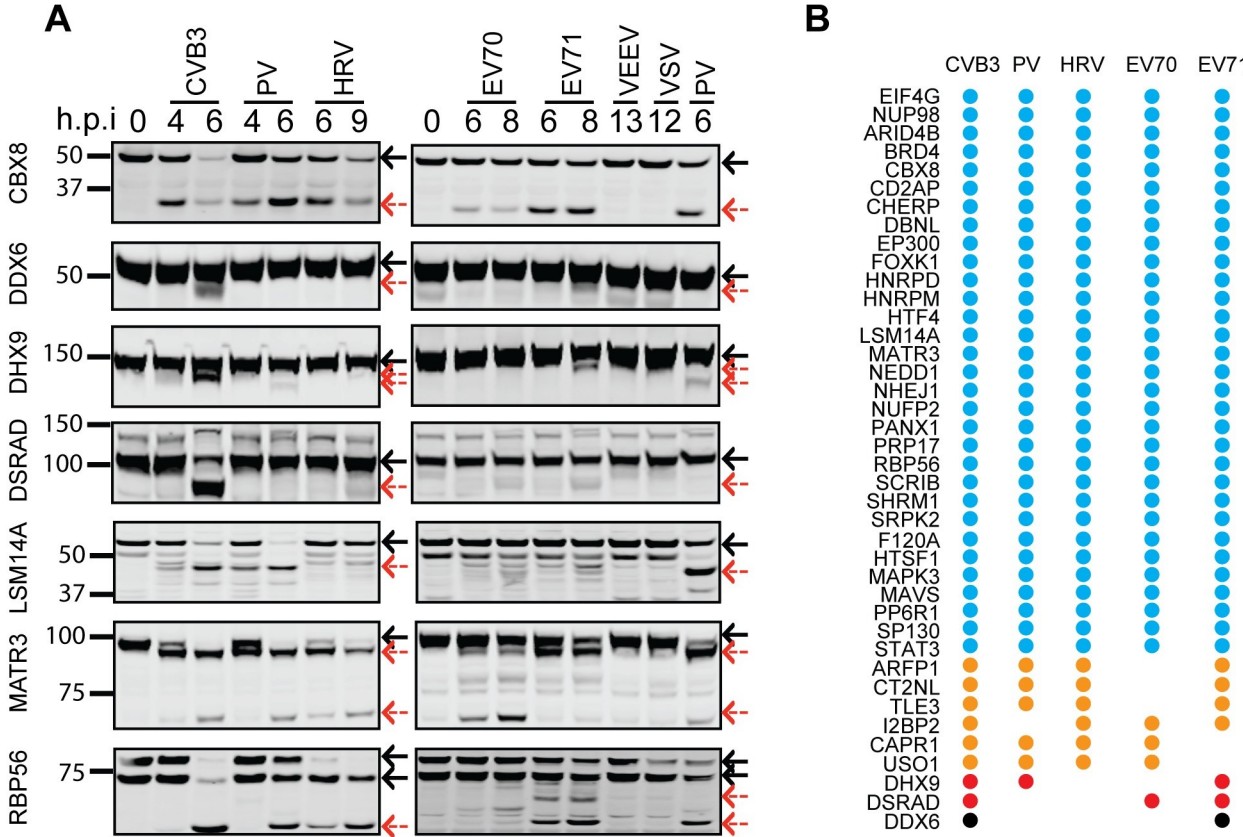

**Fig 5. Validation of proteins targeted by multiple enteroviruses.** (A) HeLa cells infected with CVB3 (eGFP-Coxsackievirus B3), PV (poliovirus type 1), HRV (human rhinovirus A16), EV70 (enterovirus D-70), EV71 (enterovirus A-71), VEEV (Venezuelan equine encephalitis virus) or VSV (vesicular stomatitis virus) were lysed at the indicated times, and equal amounts of total protein were subjected to western blot with the indicated antibodies. The black solid and red dotted arrows indicate the full-length protein and the cleavage products, respectively. (B) The western blot results are summarized to show proteins commonly targeted by all five enteroviruses and those unique to a subset of viruses (the western blot images are shown in S9 Fig).

## LSM14A is a novel innate immune response target cleaved and inactivated by enterovirus proteases

LSM14A was identified in our analysis as a protein broadly targeted by enteroviruses in a variety of cell types (Fig 5 and S10 Fig) and mature neurons (S11 Fig). This protein mainly resides in *p* bodies, cytoplasmic ribonucleoprotein granules involved in mRNA turnover, and has been previously implicated in antiviral immunity against some DNA and RNA viruses [22]. Our studies indicated that LSM14A cleavage is mediated by 2A$^{pro}$ (Fig 3) at a recognition site with a P1´ G. To validate this target site, we changed the P1´ glycine (G147) to either alanine (A) or glutamate (E) and tested if these mutants were cleaved upon virus infection (Fig 6A). Intriguingly, the G147A and G147E proteins were still cleaved, albeit less efficiently. Also, the size of the cleaved product was larger than with the wild-type protein, suggesting that the mutant proteins were cleaved at an alternate, upstream site (Fig 6B).

Next, we performed functional assays to test if LSM14A cleavage renders the protein non-functional. LSM14A has been reported to enhance innate immunity upon Sendai virus (SeV) infection [22]. To confirm this, we expressed LSM14A in 293T cells containing reporter plasmids ISRE-FLuc and RL-RLuc and infected these cells with SeV. SeV infection should activate IRF3/7, which binds to the Interferon-Stimulated Response Element (ISRE) in the ISRE-FLuc

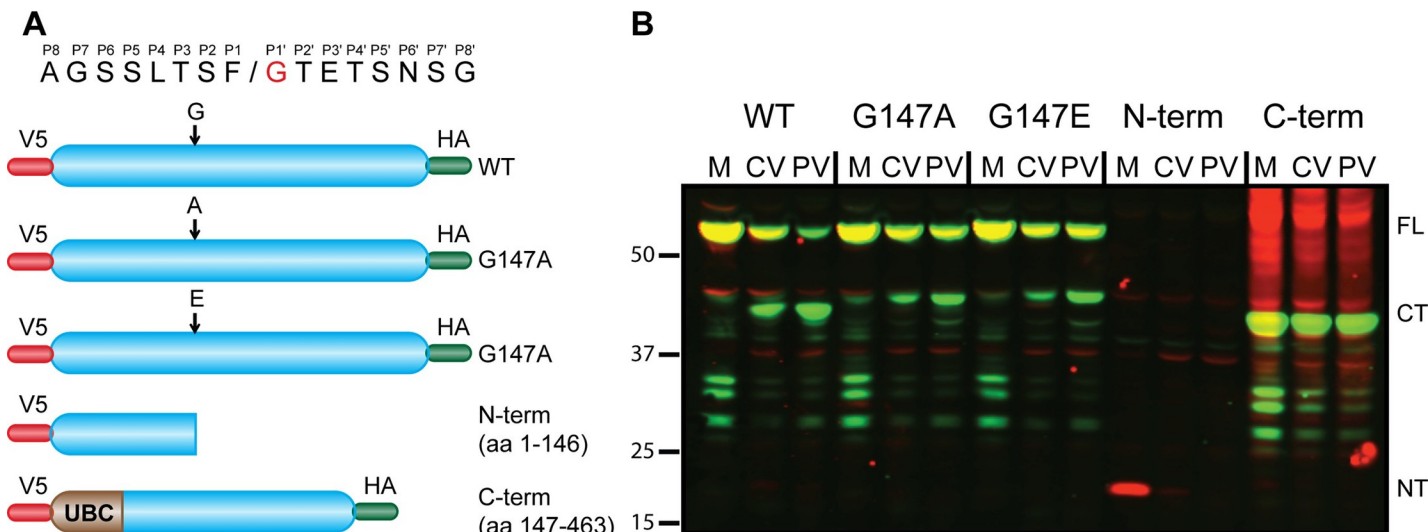

**Fig 6. Validation of LSM14A cleavage.** (A) Schematics of the LSM14A constructs generated for validation of the cleavage site. The cleavage site and the flanking eight amino acids are shown in the cartoon. The P1 glycine that was mutated is shown in red. (B) HeLa cells transduced to stably express the LSM14A constructs were infected with eGFP/CVB3 or PV, and the LSM14A cleavage was analyzed by western blot. The blots were probed with mouse anti-V5 and rabbit anti-HA antibodies followed by detection with IRDye 680RD goat anti-mouse IgG (red channel) and IRDye 800CW goat anti-rabbit IgG (green channel). FL, full-length protein; CT, C-terminal cleavage product; NT, N-terminal cleavage product; M, mock; CV; eGFP/CVB3; PV, poliovirus; UBC, Ubiquitin C.

reporter to promote Firefly luciferase (FLuc) expression. As a comparator, Renilla luciferase (Rluc) is expressed from RL-RLuc under the control of the CMV promoter, and its levels should remain stable. As expected, expression of LSM14A enhanced SeV-mediated ISRE activation in a dose-dependent manner, confirming the role of this protein in innate immunity (Fig 7A: Right panel). Interestingly, however, LSM14A exhibited no activity in the absence of SeV (Fig 7A: Left panel). This was unlike another innate immune signaling factor, MAVS (S12 Fig), and raises the possibility that LSM14A has a role in RNA sensing.

We took two approaches to test how the enterovirus-mediated cleavage affects LSM14A activity. First, we separately expressed the cleavage products in 293T cells and tested their ability to enhance SeV-induced ISRE activation. As might be expected, the individual LSM14A cleavage products did not enhance the ISRE activity (Fig 7B). In a second approach, we engineered a TEV protease (TEV$^{pro}$) recognition site into LSM14A at the location of the enterovirus 2A$^{pro}$ cleavage. We then tested ISRE activation in the presence or absence of TEV$^{pro}$. Importantly, the modified LSM14A showed comparable ISRE-FLuc activation as its wild-type counterpart in 293T cells, indicating that the insertion of the TEV$^{pro}$ cleavage site did not disrupt its function (S13A Fig). In the presence of TEV$^{pro}$, however, the modified LSM14A underwent cleavage (S13B Fig) and could no longer enhance SeV-mediated immune activation (Fig 7C). These results clearly indicated that the N- and C-terminal cleavage products of LSM14A do not cause ISRE activation upon SeV infection.

Exactly how LSM14A contributes to antiviral immunity is unknown. To gain insights into LSM14A mechanism, we examined its ISRE-activating potential in 293T cells lacking various components of antiviral immune signaling. Absence of MAVS abrogated the ability of LSM14A to enhance SeV-mediated immune signaling, suggesting that LSM14A functions upstream of MAVS (Fig 7D). Depletion of RIG-I (a known SeV sensor) also blocked LSM14A activity, indicating that LSM14A might function as a RIG-I co-factor or alternatively participate in signaling only once the pathway is activated. Finally, LSM14A activity was unaffected when STAT1 was knocked out (Fig 7D), implying that the protein does not work through

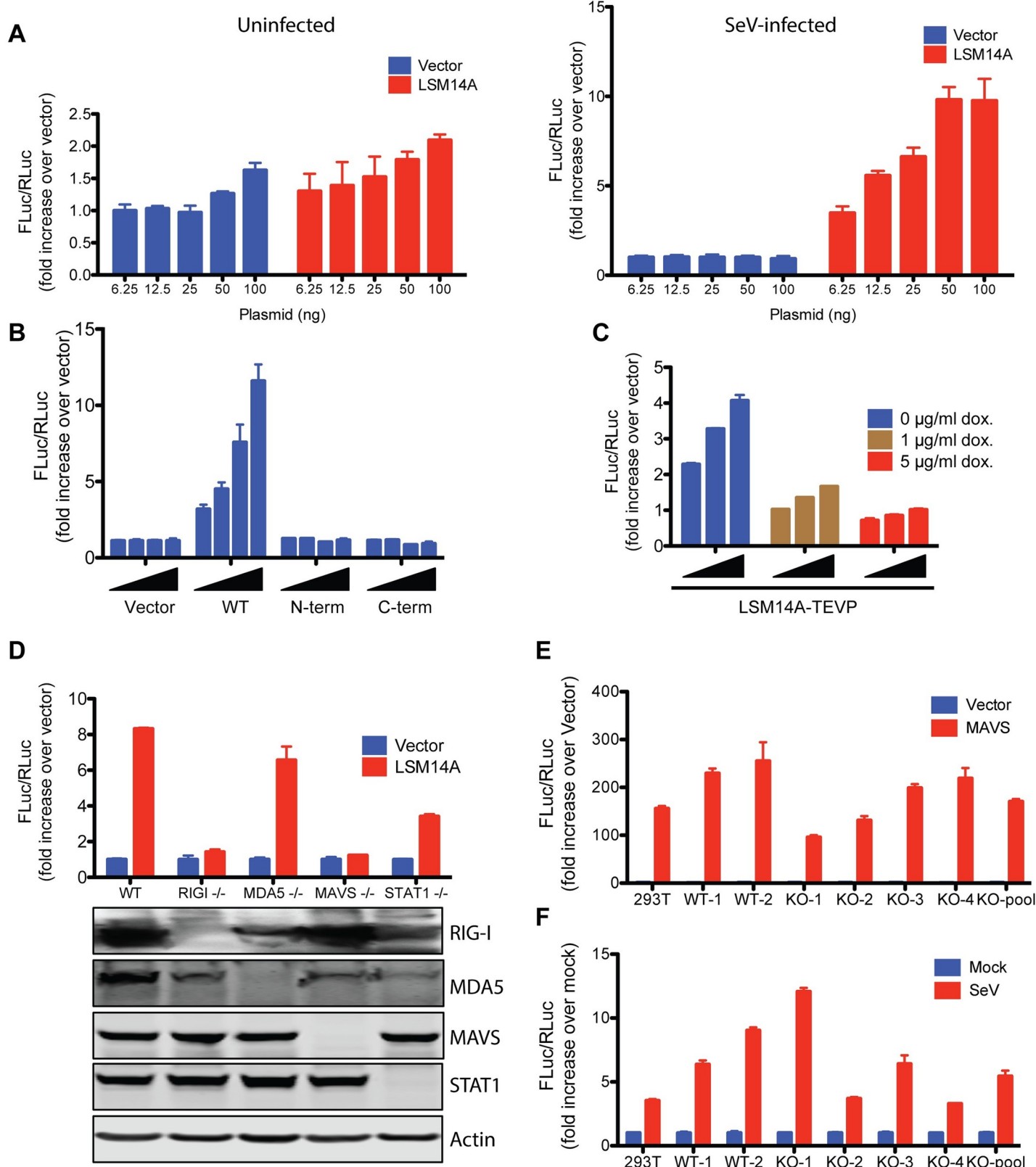

**Fig 7. Functional assays of LSM14A.** (A) LSM14A activated ISRE in a dose-dependent manner. The 293T cells (30,000 cells per well in 48-well plates) expressing the ISRE-FLuc and RL-RLuc reporters were transfected with the indicated amounts of GFP (vector) or LSM14A plasmids. The next day, the cells were left uninfected (left

panel) or infected with SeV (right panel) for 24h followed by luciferase assay. The FLuc/RLuc ratio was calculated and plotted as fold-increase over the minimum amount of vector plasmid used (6.25 ng). (B) LSM14A cleavage fragments do not activate ISRE. The reporter assays with the indicated LSM14A mutants were performed similarly as in A. The DNA concentrations of 6.25, 12, 25, and 50 ng/well were used. (C) TEVP-mediated cleavage of LSM14A blocks its ability to activate ISRE. The 293T cells containing dox-inducible TEVP and expressing the ISRE-FLuc and RL-RLuc reporters were transfected with 12.5 ng/well, 25 ng/well, or 50 ng/well LSM14A-TEVP plasmid followed by SeV infection. The indicated amount of doxycycline was added to the culture medium 24h prior to DNA transfection and kept for the entire experiment. The reporter assay was performed as described in A. (D) LSM14A requires RIG-I and MAVS for its function. The 293T cells knocked out for the indicated genes were used for the reporter assays as described in A (top panel). 50 ng/well DNA was used. The cells were tested for the expression of the indicated proteins. For detection of RIG-I and MDA5, the cells were exposed to 5 nM IFN-2a alpha for 48h. (E) MAVS does not require LSM14A for its innate immune function. The 293T cell clones expressing or lacking LSM14A were transfected with 12.5 ng/well MAVS and tested for the activation of ISRE as in A. This experiment was performed in the absence of SeV infection. KO-pool was generated by mixing the 293T cell clones that were confirmed for the loss of LSM14A. (F) SeV can activate ISRE in the absence of LSM14A. The 293T cell clones used in E were infected with SeV and the ISRE reporter activity was measured as in A. For all experiments, the data are presented as mean +/- standard deviation for triplicate samples.

interferon signaling. To further investigate whether LSM14A functions upstream of MAVS, we generated LSM14A knockout cells and examined the ability of MAVS to induce immune signaling in this setting. MAVS activity remained largely unchanged upon LSM14A depletion, supporting the hypothesis that MAVS acts downstream of LSM14A (Fig 7E). To examine the possibility that LSM14A is an essential RIG-I co-factor, we tested SeV-induced immune signaling in LSM14A knockout cells. Interestingly, LSM14A depletion did not suppress SeV-mediated ISRE activation (Fig 7F), precluding the role of LSM14A as a critical RIG-I cofactor. Taken together, it appears that LSM14A promotes but is not essential for RIG-I signaling in 293T cells. Cleavage of this protein, along with cleavage of MAVS, may help enteroviruses evade host defenses.

## Discussion

Despite mounting evidence that enteroviruses employ proteolysis to modulate or usurp host machinery, the true extent of proteolysis in a virus-infected cell remains unknown. Here, we took a relatively unbiased N-terminomics approach to obtain the first comprehensive catalog of cellular proteins that undergo cleavage upon enterovirus infection. This led to the identification of a number of known and novel substrates from diverse cellular pathways. As expected from the structural and functional similarities of enteroviral proteases [1], many of the identified proteins were targeted by all viruses tested. However, some interesting differences in the cleavage patterns emerged. For example, as demonstrated by both N-terminal labeling and western blot, DDX6 and DSRAD were efficiently cleaved only in the CVB3-infected cells. Similarly, NUP98 and DHX9 appeared to be targeted with different efficiencies by different viruses, while MAVS and F120A were cleaved by all viruses but at varying sites. Importantly, these patterns, although originally identified in HeLa cells, were seen in all cell types tested, indicating that they reflect the true virus-host biology instead of cell culture artifacts. To investigate the relevance of the cleaved proteins, we followed up on one candidate, LSM14A, and showed that the truncated forms of this protein generated by enterovirus 2A$^{pro}$-mediated cleavage do not cause ISRE activation in response to SeV infection.

LSM14A, originally identified in 1998 as an mRNA-binding protein in salamander oocytes [28], was implicated in a 2012 report in innate immune signaling upon RNA and DNA virus infections [22]. Using SeV as an example of an RNA virus and herpes simplex virus-1 (HSV-1) as an example of a DNA virus, the 2012 study showed that LSM14A enhanced ISRE activity in response to virus infection. Consistent with this, we observed a dose-dependent effect of LSM14A on SeV-induced ISRE activation in 293T cells. However, when LSM14A is cleaved by 2A$^{pro}$, it can no longer mediate SeV-induced ISRE activation, indicating a loss of function. LSM14A is a 463 aa protein with an N-terminal LSm domain of 76 aa (aa 1–76) followed by an intrinsically unstructured region of around 200 aa and then two C-terminal motifs, namely a DFDF box (aa 291–316) and an FDF_TFG box (aa 361–397) [22]. The C-terminal region of

the protein is essential for RNA-binding and *p* body localization [22, 29], two characteristics that are believed to be important for LSM14A antiviral immune activity. This region alone however cannot mediate immune signaling and requires the presence of the N-terminal LSm domain for activity. In line with this, when the N- and C-terminal domains of LSM14A are separated by enteroviral cleavage, the protein becomes inactive. Unfortunately, we could not show the direct relevance of LSM14A to enterovirus infection due to several biological and technical limitations, such as the robustness of virus infection in most cell lines, redundancy in the innate immune pathways, and cleavage of several other immune components by this group of viruses. For instance, enteroviruses are known to target RIG-I [30] and MAVS, the two proteins our knockout experiments suggested are essential for LSM14A activity (Fig 7D). This diminished the utility of typical overexpression and knockout strategies to decipher the function of LSM14A during enterovirus infection.

The N-terminomics approaches simultaneously reveal the identity of the proteins being cleaved and the exact site within a protein where the cleavage occurs. This allowed us to search for proteins directly targeted by enteroviral proteases. We first generated a list of candidates that were found cleaved in at least two of the three CVB3-infected replicates but in no uninfected replicate and then cross-referenced this list with the consensus sequence motifs that enteroviral proteases are known to target. This yielded a number of proteins from various cellular processes, many of which have been previously implicated in enterovirus replication. For example, proteins essential for various steps during host gene expression, autophagy, and antiviral immune signaling were identified. Similarly, several components of the cytoskeleton, cytoplasmic ribonucleoprotein granules, and the nuclear pore complex were among the proteins targeted for cleavage. Enteroviruses are well known for reducing new protein synthesis by translational shut off early in infection, as confirmed by the cleavage of eIF4G at just 2 h.p.i. Our finding that proteolysis targets numerous additional factors involved in gene expression emphasizes the way these viruses remodel the cell for their benefit. Importantly, western blot analysis of infected cells confirmed all the cleavages tested, and the *in vitro* assay further validated that the cleavages were mediated by the viral proteases. It should however be noted that the *in vitro* assay does not completely rule out the possibility of a cleavage being mediated by a host protease. It is conceivable that mixing an enteroviral protease with the HeLa cell lysate stimulated a host protease that then targeted its substrates.

As the virus-infected cells approached complete cytotoxicity, the numbers of cleaved proteins substantially increased. This was likely the cumulative effect of the high levels of viral proteases per cell and the loss of subcellular compartmentalization, allowing the viral proteases to contact more cellular proteins. Also, extensive changes in the intracellular milieu might result in the activation of host proteases, and indeed we observed a surge of caspase activity as CPE became evident. Inferring that late-stage cleavages are likely to be non-specific, we included only those proteins cleaved early in infection when listing high-confidence targets. It is worth noting, however, that several functionally important cleavages have been reported to occur late in enterovirus infection. For example, many proteins that enteroviruses recruit at early stages of infection but need to eliminate at later stages are removed by protease-mediated cleavage [25]. Also, whereas in most cases a cleavage renders the substrate non-functional, there are instances where a cleavage endows the protein with a new function beneficial for the virus. For example, RIP3, a kinase that regulates autophagy and orchestrates necrotic cell death, is an essential host factor for the early phase of CVB3 replication. Later in infection, RIP3 is cleaved by CVB3 3C$^{pro}$, which ablates its ability to mediate necrotic cell death and generates a C-terminal cleavage product that is utilized by CVB3 to induce a non-necrotic form of cell death [25]. While we focused on early cleavages in this study, understanding late stage cleavages could be equally important.

The advent of various N-terminomics approaches has spurred interest in the role of viral proteases in virus-host interactions. Recently, Jagdeo et al., used TAILS, another N-terminomics approach, to inventory cellular proteins cleaved by the enteroviral 3C$^{pro}$ [21]. They mixed HeLa or HL-1 (murine cardiac muscle cells) cell lysates with the purified viral protease from CVB3 or PV and identified proteins that were cleaved by the wild-type but not catalytically inactive 3C$^{pro}$. This yielded a list of 34 high-confidence substrates, seven of which the authors subsequently validated in virus-infected cells. Interestingly, six of these seven substrates were present in our dataset, and importantly, for one substrate, RIPK1, where the authors saw discrepancy between the TAILS and western blot results, the subtiligase method we employed identified the cleavage site that matched the products observed in western blot. This emphasizes the importance of applying N-terminomics approaches on virus-infected cells rather than protease-mixed cell lysates, as the former is likely to yield more accurate and complete information about host targets and sites of cleavage. Another advantage of using a virus infection system is that it allows for the identification of temporally and spatially regulated cleavage events.

In all, this report provides the first comprehensive view of proteolysis in enterovirus-infected cells and adds LSM14A to the growing list of host proteins that enteroviruses target for cleavage to generate a virus-friendly environment. The inventory of candidate proteins presented in this study provides a foundation for future investigations into the complex interplay between enteroviruses and their host cells, and the approach and pipeline can be applied to define the proteolysis landscape of any virus-host encounter.

## Materials and methods

### Cells and viruses

Human cervical carcinoma HeLa cells (ATCC CCL-2), human embryonic kidney HEK293T cells (ATCC CRL-3216), human muscle rhabdomyosarcoma RD cells (ATCC CCL-136), and human neuroblastoma SK-N-SH cells (ATCC HTB-11) were maintained in Dulbecco's minimum essential medium (DMEM) supplemented with 10% fetal bovine serum (FBS). Human colorectal adenocarcinoma Caco-2 cells (ATCC HTB-37) were grown in DMEM containing 20% FBS. 293T cells deficient for RIG-I, MDA5, MAVS, and STAT1 have been previously described and were kindly provided by Veit Hornung (Universitat Bonn, Germany) [31].

For neural progenitor cells (NPC), we first differentiated the human embryonic stem cells (hESC) into neuroectoderm cells using the dual SMAD inhibition method (Chambers et al., Nature Biotechnology. 2009). Briefly, hESC were dissociated into single cells using Accutase and plated onto Matrigel-coated plates in the presence of 10 μM ROCK inhibitor (Y-27632) to achieve the confluence of approximately 90% the next day. Differentiation was induced with SRM (Knockout DMEM/F12 containing 15% knockout serum replacement (KOSR) and 1% GlutaMax) supplemented with 10μM SB431542 and 200nM Noggin for five to seven days. Confluent ectoderm cells were passaged and maintained for one more week in the induction medium containing dual SMAD inhibitors. Derived NPC were then maintained in neural progenitor medium (Knockout DMEM/F12 containing 2% StemPro Neural Supplement and 1% GlutaMax) supplemented with 20 ng/ml bFGF and 20 ng/ml EGF. NPC cultures were fed fresh medium daily. The reagents used for stem cell differentiation were obtained from the following vendors: Matrigel (# 356230), Corning; ROCK inhibitor Y-27632 (# 72308), STEMCELL Technologies; KOSR (# 10828028), Life Technologies; GlutaMax (# 35050–061), Life Technologies: SB431542 (# 1614), Tocris Bioscience; Noggin (# 120-10C), Peprotech; StemPro Neural Supplement (# A10508-01), Life Technologies; bFGF AA 10–155 (# PHG0024), Life Technologies; animal-free human EGF (# AF-100-015), Peprotech; and knockout DMEM/F-12 (# A1370801), Life Technologies.

With the exception of EV71, all enteroviruses used in this study were rescued from infectious cDNA clones obtained from various investigators: pMKS1-GFP [32] (referred to as eGFP/CVB3 in this paper) was obtained from Dr. J. Lindsay Whitton of the Scripps Research Institute, La Jolla; poliovirus (pT7-Manony) [33] and human enterovirus 70 (pDNE9) [34] from Dr. Vincent Racaniello of Columbia University, New York; and human rhinovirus A16 strain (pA16) [35] from Dr. Ann Palmenberg of the University of Wisconsin, Madison. Human enterovirus 71 was obtained from BEI resources (cat. no. NR-471). The Cantell strain of the Sendai virus (SeV) used for innate immune induction experiments was generated in the laboratory of Dr. Adolfo Garcia-Sastre of The Icahn School of Medicine at Mount Sinai, New York, by inoculation into the allantoic cavity of 10-day-old embryonated chicken eggs. Following incubation at 37˚C for 48h, allantoic fluid was harvested and titrated by hemagglutination of chicken red blood cells. GFP-tagged version of the VEEV vaccine strain TC83 (VEEV-GFP) [36] was from Ilya Frolov of the University of Texas, Galveston, and the GFP-tagged VSV (VSV-GFP) [37] was from John Rose of the Yale University, New Haven.

## Virus infections

*Large-scale experiments*: For each infection experiment, HeLa cells were plated into twenty-four 150-cm dishes (six dishes for each of the four time points) at the density of 10 million cells per dish. The next day, we infected the cells with the virus at a multiplicity of infection (m.o.i.) of 5 plaque-forming units (PFU)/cell. Since eGFP/CVB3 did not yield plaques, we could not calculate m.o.i. for this virus. Therefore, we optimized the infection conditions for eGFP/CVB3 using different dilutions of the virus stock and selecting the dilution that yielded 100% GFP-positive cells at 4 h.p.i. For infection, the virus was diluted in 5 ml of opti-MEM and adsorbed to the cell monolayer at 37˚C for 1h. The virus inoculum was then removed and 25 ml of DMEM/10% FBS was added to each dish. The cells were harvested at the indicated times determined by prior small-scale infection experiments described below. For each time point, we divided six dishes into three sets of two dishes each and pooled the cells from each set to obtain the total of three replicates. The uninfected cells were harvested concomitant with the last infection time point. To harvest, we scraped the cells into enzyme-free cell dissociation solution (Millipore: #S-014-C) and pelleted them by centrifugation at 1,000 x*g* for 5 min. The cell pellets were then flash frozen in liquid nitrogen and stored at -80˚C.

*Small-scale experiments*: To determine the optimal m.o.i. and times of harvest for large-scale experiments, we seeded HeLa cells into 24-well plates at a density of 175,000 cells per well and allowed them to grow overnight at 37˚C. We infected the cells at the m.o.i. of 1, 5, or 10 by first diluting the virus in 100 μl of opti-MEM and then incubating the cells with this inoculum at 37˚C for 1h. The virus inoculum was then removed and fresh DMEM/10% FBS medium was added. The cells were then fixed at various times after infection with 4% paraformaldehyde (PFA) and stained with antibodies against viral antigens.

## Antibodies and chemicals

The antibodies used for immunofluorescence include anti-PV 1 antibody clone 583-G8-G2-A4 (Millipore: #MAB8560), anti-EV-D70 antibody clone 74-5G (Millipore; #MAB843), anti-EV-A71 antibody (GeneTex; #GTX132339), and anti-rhinovirus antibody clone R16-7 (LifeSpan BioSciences; #LS-C200976), anti-LSM14A antibody (Bethyl Laboratories; #A305-103A). Most antibodies for western blot were purchased from Bethyl Laboratories. These included rabbit polyclonal antibodies against ARFP1 (#A304-676A), AR14B (#A302-233A), CAPR1 (#A303-882A), CBX8 (#A300-882A), CD2AP (#A304-728A), CHERP (#A304-621A), DBNL (#A303-351A), DDX6 (#A300-461A), DHX9 (#A300-855A), EP300 (#A300-

358A), F120A (#A303-889A), HNRPM (#A303-910A), HTF4 (#A300-754A), HTSF1 (#A302-023A), I2BP2 (#A303-190A), LSM14A (#A305-103A), MAPK3 (#A304-305A), MATR3 (#A300-591A), MAVS (#A300-782A), NEDD1 (#A304-545A), NHEJ1 (#A300-730A), NUFP2 (#A301-600A), NUP98 (#A301-786A), PP6R1 (#A300-968A), PRP17 (#A303-700A), RBP56 (#A300-309A), SP130 (#A302-491A), and SRPK2 (#A302-467A). The following antibodies were purchased from Proteintech: CT2NL (#25523-1-AP), PANX1 (#12595-1-AP), SHRM1 (#18218-1-AP), TLE3 (#11372-1-AP), and USO1 (#13509-1-AP).

The antibodies purchased from the Cell Signaling Technology include anti-BRD4 rabbit monoclonal (#13440), anti-eIF4G rabbit polyclonal (#2498), anti-FOXK1 rabbit polyclonal (#12025), anti-HNRPD rabbit monoclonal (#12382), and anti-STAT3 rabbit polyclonal (#9132). Additional primary antibodies include anti-DSRAD mouse monoclonal (Santa Cruz Biotechnology; #sc271854), anti-SCRIB rabbit polyclonal (GeneTex; #GTX107692), anti-HA rabbit polyclonal (Abcam; #ab9110), anti-V5 mouse monoclonal (ThermoFisher Scientific; #MA5-15253), and IRDye 800CW-conjugated anti-streptavidin antibody (Rockland Immuno-chemicals; S000-31). The secondary antibodies used for western blot included IRDye 680RD goat anti-mouse (LI-COR; 926–68070) and IRDye 800CW goat anti-rabbit (LI-COR; 926–32211).

The following chemicals were purchased from Sigma: Acetonitrile (#271004), TCEP (#75259), iodoacetamide (I1149), AEBSF (#A8456), and PMSF (#P7626). E-64 protease inhibitor (#324890) and Z-VAD pan-caspase inhibitor (#219007) were purchased from EMD Millipore. Sequencing grade trypsin was purchased from Promega (#V5113) and Lysyl endopeptidase from Wako (#125–05061).

## Plasmids

LSM14A cDNA was PCR amplified from the total HeLa cell cDNA and appended at its 5' and 3' ends with the V5 and HA tags, respectively. The PCR primers were engineered to introduce a BamH1 cleavage site upstream of the V5 tag and an Nhe1 cleavage site downstream of the HA tag. Using these cleavage sites, we cloned LSM14A cDNA into the pLOC lentiviral vector and pcDNA3.1 expression plasmid to obtain pLOC-LSM14A and pcDNA3.1-LSM14A, respectively. For LSM14A cleavage-defective mutants, we replaced glycine at position 147 with alanine or glutamic acid by overlapping PCR. The N-terminal V5-tagged ubiquitin (Ubi) was fused with amino acid 147–463 of LSM14A to obtain pV5-Ubi-LSM14A. The pISRE-FLuc plasmid containing ISRE-driven Firefly luciferase reporter was from Stratagene (#219089) and pRL-RLuc, containing CMV promoter-driven Renilla luciferase reporter, was generated by modifying the pRL-HL plasmid, generously provided by Dr. Stanley Lemon [38]. The small guide RNA (sgRNA) plasmid pSpCas9(BB)-2A-Puro (PX459) (Addgene # 481239) was a gift from Feng Zhang. We designed three sgRNA for CRISPR knockout of LSM14A using crispor.tefor.net: sgRNA1 (exon1), 5'-AGCGGGGGGCACCCCTTACAT-3'; sgRNA2 (exon 2), 5'-GGTGGGTA TTGGACGATCTGT-3'; and sgRNA3 (exon 4), 5'-TTACCCCAAAGTAGTGCGGT-3'. The sgRNA were then cloned into PX459 plasmid for subsequent transfection into cells.

## Production and purification of recombinant proteins

**TEV protease.** The recombinant TEV protease was produced in *Escherichia coli* as previously described [39]. Briefly, a His$_6$-tagged TEV S219V mutant was cloned into pET vector and transformed into BL21 DE3 cells. The next day, we inoculated one bacterial colony into 50 ml of LB medium and grew overnight at 37˚C. Twenty-four hours later, 2.5 ml from the overnight bacterial culture was transferred into 1L of fresh LB medium and the cells were grown in a shaking incubator at 37˚C until OD$_{600}$ reached ~ 0.5. The cells were chilled on ice, and the

protein expression was induced with 1mM IPTG. After shaking at 30°C for 4h, the cells were harvested by centrifugation. Cell pellets were resuspended in the storage buffer (20% sucrose and 20 mM Tris pH 8), flash frozen in liquid nitrogen, and stored at -80°C.

To purify the TEV protease, the cells were lysed by sonication in the presence of 500 mM NaCl, 20 mM Na/K phosphate, 1 mM PMSF, 0.5% IGEPAL (octyl-phenyl-polyethylene glycol), 15 mM imidazole, 1 mM ß-ME (ß-mercaptoethanol), 50 µg/ml lysozyme, 2 µg/ml DNase I, and 2 µg/ml RNase A. The lysate was centrifuged at 40,000 x$g$ at 4°C for 1h to remove the insoluble material. The protein contained in the supernatant was captured on Ni-NTA agarose beads (Qiagen) by incubating the supernatant with the beads for 1h at 4°C. The beads were then loaded on an Econo-Pac column (Bio-Rad) and washed with 25 column volumes of washing buffer (500 mM NaCl, 20 mM Na/K phosphate, 15 mM imidazole, 20 mM Tris pH 8, and 1 mM ß-mE). The bound protein was eluted off of the beads with elution buffer (500 mM NaCl, 20 mM Na/K phosphate, 250 mM imidazole, 20 mM Tris pH8, and 1 mM ß-ME) and further purified by gel-filtration using a HiPrep 26/60 Sephacryl S-200 HR column (GE Healthcare).

**Subtiligase.** *Bacillus subtilis* BG2864 bacteria, containing a plasmid carrying subtiligase, were grown overnight at 37°C in 2X TY medium containing 12.5 µg/ml chloramphenicol. The culture was centrifuged at 4500 x$g$ for 15 min at 4°C and the bacterial pellet was discarded. The supernatant was placed on a magnetic stirrer and the protein in the supernatant was precipitated by slow addition of ammonium sulfate at 4°C. After 1h of mixing, the protein was pelleted by centrifugation at 10,000 x$g$ for 30 min at 4°C. The pellet was dissolved in a buffer containing 5 mM DTT and 25 mM sodium acetate pH 5.0 followed by addition of three volumes of ethanol to obtain the final concentration of 75%. Following another 30-min mixing on a stir plate, the solution was centrifuged at 5000 x$g$ for 15 min at 4°C. The protein pellet was resuspended in the above buffer (5 mM DTT and 25 mM sodium acetate pH 5.0) and dialyzed against 9L of the same buffer overnight at 4°C. After dialysis, the protein was sequentially purified with ion exchange and gel filtration columns (HiTrap SP HP 5 ml and Superdex 200 10/300GL; GE Healthcare). The concentration of the purified protein was set at 100 µM and stored at -80°C.

**CVB3 proteases.** For wild-type 2A$^{pro}$, wild-type 3C$^{pro}$, and C147A 3C$^{pro}$, the BL21 (DE3) T1R pRARE2 bacteria containing the plasmids with His-tagged protease constructs were grown overnight in the presence of 100 µg/ml kanamycin and 34 µg/ml chloramphenicol. When OD$_{600}$ reached 0.3, the protein expression was induced by overnight treatment with 0.5 mM IPTG. For 2A$^{pro}$, we used autoinduction media (Thermo: K6803) and also added 1 µM ZnCl$_2$ to facilitate protein folding. The cultures were then centrifuged at 4500 x$g$ for 10 min and the pellets were lysed by pulse sonication in the presence of 100 mM HEPES, 500 mM NaCl, 10% glycerol, 10 mM imidazole, and 0.5 mM TCEP, pH 8.0. The lysates were centrifuged at 49,000 x$g$ for 20 min and the supernatants were filtered through 0.45 µm filters. The samples were then loaded onto ÄKTA Xpress (GE Healthcare) and purified first with an IMAC column 5 ml HisTrap HP (GE Healthcare) and then with a gel filtration column HiLoad 16/60 Superdex 75 (GE Healthcare). As our constructs contained the TEV protease cleavage site cloned between the His tag and the N-terminal end of the CVB3 proteases, we removed the tag by incubating the samples with TEV protease at a protease-to-protein ratio of 1:25 at room temperature for 2h. The cleaved His tag was then removed from the protein preps by passing the samples through the HisTrap HP column. The flow through was collected and concentrated to 1 mg/ml in the storage buffer (25 mM HEPES pH 7.5, 140 mM NaCl, 2 mM TCEP, 10% glycerol) followed by lyophilisation and storage at -80°C.

For C110A 2A$^{pro}$, the His-tagged protease was cloned into pBVboostFGIIWPRE C43 plasmid and transformed into BL21 DE3 bacteria. The bacteria were grown in 5 mM glucose and

7 μg/ml gentamycin to the $OD_{600}$ of 0.4 and protein expression was induced by overnight treatment with 0.1 mM IPTG at 28˚C. The bacterial culture was then centrifuged at 4500 x*g* for 10 min and lysed by sonication in the binding buffer (50 mM Tris pH 8, 0.5 M NaCl, 10 mM imidazole) containing 30 μg/ml lysozyme. The cell lysate was centrifuged at 10,000 x*g* for 20 min at 4˚C and the supernatant was incubated with Ni-NTA agarose beads. The captured protein was eluted off the beads using 50 mM Tris pH 8, 0.5 M NaCl, and 500 mM imidazole followed by removal of imidazole with VivaSpin Turbo 10K centrifugal filter (Sartorius). The protein was further purified by anion exchange chromatography using Q Sepharose Fast Flow (GE Healthcare), lyophilized in the storage buffer (25 mM HEPES pH 8, 140 mM NaCl), and stored at -80˚C.

## Preparation of TEVest4

The TEVest4 peptide is an amine terminal biotinylated peptide comprised of the sequence: Biotin-eAhx-eLys-Gly-Gly-Thr-Glu-Asn-Leu-Tyr-Phe-Gln-Abu-Glc-Arg-NH2. The peptide was synthesized by solid-state peptide synthesis with slight modifications from the original protocol [40]

## N-terminal peptide isolation

For labeling, the frozen cell pellets were lysed by sonication in a buffer containing 1% SDS, 100 mM bicine pH8.0, 5 mM EDTA, 50 μM z-VAD-fmk, 50 μM E-64, 500 μM AEBSF, and 50 μM PMSF. The lysates were first passed through QIAshredders (Qiagen; #79654) to shred the remaining nucleic acids and then through detergent removal columns (ThermoFisher Scientific; #87777) to remove SDS. The total cellular protein in all samples was set at 15 μg/μl to obtain a total of 3 mg protein in a 200-μl volume. The lysates were incubated at room temperature for 1h with a biotinylated peptide ester (TEVest4; 10 mM stock in DMSO) and purified recombinant subtiligase (100 mM stock; see below) to obtain the final concentrations of 1 mM and 1 μM, respectively. The unincorporated peptide esters were removed by acetonitrile precipitation.

The air-dried protein pellets were then dissolved in 6M guanidine hydrochloride (GnHCl) containing 100 mM bicine pH 8.0 and 10 mM TCEP. The samples were incubated at 95˚C until the pellets were completely dissolved. This was followed by an alkylation reaction, where the samples were incubated with 10 mM iodoacetamide (IAM) at room temperature for 45 min in the dark. The IAM reaction was then quenched with 10 mM DTT and biotinylated proteins were captured overnight on immobilized NeutrAvidin agarose beads (ThermoFisher Scientific; #29202).

The beads were washed five times with a wash buffer containing 4M GnHCl and 100 mM bicine, and digested overnight at 37˚C in digest buffer (100 mM ammonium bicarbonate, 1M GnHCl, 100 mM NaCl, 5 mM $CaCl_2$) containing a mixture of 3 μg sequence-grade trypsin and 1.5 μg lysyl endopeptidase. Following digestion, the beads were washed five times with wash buffer and resuspended in TEV protease buffer containing 100 mM ammonium bicarbonate, 1 mM EDTA, and 2 mM DTT. Peptides were released by digestion with 40 μg TEV protease for 6h at room temperature and desalted with C18 Empore (3M) StAGE tips, prepared in house [41].

## LC-MS/MS

We dried the desalted samples and solubilized in 2% acetonitrile and 2% formic acid prior to analysis by one-dimensional reversed-phase nano-LC-MS/MS (Q-Exactive Plus, Thermo-Fisher Scientific). For reversed-phase liquid chromatography (LC), we used $C_{18}$ columns of 12

cm length and 75 μm internal diameter packed with beads of 3 μm particle size (Nikkyo Technologies, Japan), and delivered an 80 min gradient increasing from 5% B/95% A (A: 0.1% formic acid in water, B: 0.1% formic acid in 80% acetonitrile) to 40% B/60% A at a flow rate of 300 nL/min (Dionex 3000, ThermoFisher Scientific). Peptides were concentrated using a trap column (ThermoFisher Scientific, #164564-CMD) prior to separation. MS1 was scanned from m/z 300 to m/z 1400. For HCD MS/MS acquisition, a resolution of 17,500 was used with Auto Gain Control of $5e^5$ and a maximum injection time of 100 ms. The lowest mass was set at m/z 140. For the Q-Exactive Plus mass spectrometer, we subjected the 20 most abundant multiply charged (2+, 3+, 4+, and 5+) peptides to tandem MS. Primary data will be made available upon request.

### Interpretation of MS/MS spectra

LC-MS/MS data were analyzed using Proteome Discoverer 2.3.0.523 software combined with MASCOT v. 2.3. Tandem MS data were queried against the UniProt human proteome (September 2018 release) concatenated with common contaminants and virus proteome sequences. The minimal peptide length was set at 7 amino acids. The filter of 5% false discovery rate (FDR), calculated by Percolator [42], was applied. Semi-tryptic digestion search constraints were used. All cysteines were treated as being carbamidomethylated. Methionine oxidation and N-terminal C4H7NO (α-aminobutyric acid, Abu), the latter resulting in a mass increase of 85.052764 Da, were allowed as variable modifications. Peptide intensities and spectral counts were used for quantification.

### Bioinformatic analysis

Qualitative analysis was performed to identify proteins cleaved in virus-infected cells. For this, we first required that the peptides must be Abu-labeled, identified with high confidence in at least two of the infected replicates, and absent in uninfected cells. Next, we collected gene identifiers assigned to cleaved peptides to create lists of genes with products cleaved following infection. We excluded non-human proteins. To identify proteins that are commonly cleaved among enteroviruses, we used the following filters: 1) A peptide must be identified with high confidence for at least one virus, 2) the peptide must be present in a minimum of two infected replicates for at least two viruses, and 3) the peptide must not be detected in any uninfected replicate. To account for exopeptidase nibbling (ragging), we wrote, validated, and implemented a script in R allowing for removal of those peptides from our final analyses that presumably arose from exopeptidase activity. We then input the final list of proteins into the "Search Tool for Recurring Instances of Neighboring Genes" (STRING) at https://string-db. org [43]. Only those interactions meeting the "highest confidence" criteria in the STRING database were included in the resulting interaction network. This network was then exported and visualized in Cytoscape [44], and clusters within the network were identified using the MCODE clustering algorithm [45] as implemented in the Cytoscape plugin "clusterMaker2" [46]. We functionally profiled the clusters that contained more than five proteins using GO (gene ontology) enrichment analysis as implemented in the "clusterProfiler" R package (version 3.12.0) [47]. GO terms with FDR-adjusted enrichment $p$-values less than 0.05 were considered significant. While choosing proteins for validation by western blot, we required that the P1 and P1´ amino acids match with the enteroviral protease cleavage site specificity.

### *In vitro* cleavage assay

HeLa cells were lysed by sonication in a buffer containing 100 mM bicine pH 8.0, 140 mM NaCl, 5 mM DTT, and 0.1% Triton-X. The lysate was then centrifuged at ~21,000 x*g* for 20

min at 4˚C to pellet down the insoluble material. The protein concentration was measured by BCA assay using Pierce BCA Protein Assay Kit (ThermoFisher Scientific; # 23225) and set at 2 μg/μl. For the CVB3 2A$^{pro}$ cleavage assay, we mixed 100 μl cell lysate (200 μg protein) with 1 μg of wild-type or catalytically inactive protease to obtain a protease-to-protein ratio of 1:200. For the CVB3 3C$^{pro}$ cleavage assay, we used a protease-to-protein ratio of 1:2. These ratios were determined by pilot experiments where cell lysates and proteases were mixed in different proportions and cleavage of known substrates was used as a guide. The digestion reaction was carried out at 37˚C for 3h, followed by addition of a sample loading dye and 10 min incubation at 70˚C. Protein cleavages were tested by western blot.

### Generation of LSM14A knockout cells

293T cells were transfected with the PX459 plasmid containing LSM14A sgRNA using Lipofectamine 2000 DNA transfection reagent (ThermoFisher Scientific, #11668030). We prepared three cell populations; first, transfected with empty PX459 as a control; second, transfected with the combination of sgRNA 1 and 2; and third, with the combination of sgRNA 2 and 3. The transfected cells were exposed to 2.5 μg/ml puromycin for 48h starting at 36h post-transfection and then recovered for 3 days in puromycin-free medium. The single cell clones were obtained by seeding the cells into 96-well plates at a density of 0.7 cells per well. The resulting cell clones were screened for the expression of LSM14A by western blot.

### Luciferase assay

For dual luciferase assays, cells were seeded into 48-well plates at a density of 30,000 cells per well in 250 μl volume and grown overnight at 37˚C. The next day, the cells in each well were transfected with 50 ng of pISRE-FLuc plasmid DNA and 2.5 ng of pRL-RLuc plasmid DNA along with various concentrations of the genes of interest (for example, pcDNA3.1-LSM14A, pcDNA3.1-MAVS, or pcDNA3.1-GFP herein referred to as "vector"). XtremeGENE 9 (Roche; # 06365787001) was used as a DNA transfection reagent. The next day, the cells were infected with SeV for 24h and lysed with 1X passive lysis buffer (Promega) according to the manufacturer's recommendations. Firefly and Renilla luciferase activities were measured with the Dual-Luciferase Reporter Assay System (Promega) using a Lumat LB9507 Luminometer (EG & G Berthold, Bad Wildbad, Germany).

### Supporting information

**S1 Fig. Optimization of subtiligase labeling.** (A) SDS lysis yielded better results than Triton-X lysis. HeLa cells infected with eGFP/CVB3 were lysed in either 1% Triton-X or 1% SDS. The experiment was performed in triplicate. All SDS lysates were sonicated, while out of the three Triton-X lysates, two were sonicated and one was left unsonicated. The western blot was performed with anti-histone H3 and anti-GFP antibody. (B) SDS depletion before labeling was necessary for subtiligase reaction. The HeLa cell lysates were prepared with either 1% Triton-X (single) or 1% SDS (triplicate). Of the three SDS lysates, two were subjected to SDS removal before subtiligase labeling while one was labeled in the presence of SDS. The Triton-X lysate contained the detergent during labeling. The labeling efficiency was determined by western blot with streptavidin. In parallel, the western blot for histone H3 and GFP was also performed.
(TIF)

**S2 Fig. Subtiligase labeling of eGFP/CVB3-infected HeLa cells.** (A) HeLa cells were infected with eGFP/CVB3, and the infection was monitored by GFP detection at the indicated times.

DAPI was used for nuclear staining. The bright field (BF) images show progressive cell rounding with infection. (B) HeLa cells infected with eGFP/CVB3 for 2, 4, or 6h, or left uninfected, were lysed in 1% SDS. The lysates were then depleted of SDS and either subjected to western blot to detect GFP and actin (left panel) or incubated with subtiligase and biotinylated peptide. The labeling efficiency was determined by western blot with streptavidin. The two different exposure times are shown.
(TIF)

**S3 Fig. Cleavages at CVB3 2A$^{pro}$ motifs.** The CVB3 2A$^{pro}$ preferably cleaves proteins at Y-G, T-G, F-G, V-G, and A-G pairs. We calculated the frequency of these cleavages at different times post-infection. The results of Y-G and T-G are shown in Fig 1D, while the rest are shown here.
(TIF)

**S4 Fig. Validation of protein cleavages identified by subtiligase labeling.** HeLa cells were infected with eGFP/CVB3 and lysed at 0, 2, 3, 4, 5, and 6 h.p.i. followed by western blot analysis of the indicated proteins. An equal amount of total protein, as quantified by the BCA assay, was loaded for each time point. The GFP expression was used to monitor the progression of infection (last panel). The black solid arrows indicate the full-length protein, while the cleavage products are shown with red dotted arrows. Some of the western blot images are shown in Fig 2.
(TIF)

**S5 Fig. Validation of cleavage targets by overexpression followed by western blot.** (Top panel) General schematics of protein tagging with V5 at the N-terminal end and HA at the C-terminal end. (Bottom panel) HeLa cells transduced to stably express the indicated doubly tagged proteins were infected with eGFP/CVB3 for 4h or 6h, or left uninfected followed by detection of cleavage by western blot. The blots were probed with mouse anti-V5 and rabbit anti-HA antibodies and detected with IRDye 680RD goat anti-mouse IgG (red channel) and IRDye 800CW goat anti-rabbit IgG (green channel). The full-length protein is indicated with black solid arrows, while the cleavage products are indicated with red dotted arrows.
(TIF)

**S6 Fig. *In vitro* cleavage assay of the identified proteins.** HeLa cell lysates (200 μg protein) were incubated with CVB3 2A$^{pro}$ or its catalytically inactive mutant C110A (1 μg), or CVB3 3C$^{pro}$ or its catalytically inactive mutant C147A (100 μg) at 37˚C for 3h and analyzed by western blot. Lysates from the uninfected and eGFP/CVB3-infected HeLa cells were included as positive controls. The full-length protein is indicated with black solid arrows, while the cleavage products are indicated with red dotted arrows. Some of the western blot images are shown in Fig 3.
(TIF)

**S7 Fig. Growth kinetics of enteroviruses in HeLa cells.** The cells infected with PV (poliovirus), HRV (human rhinovirus A16), EV70 (enterovirus D-70) and EV71 (enterovirus A-71) were fixed in 4% PFA at the indicated times post-infection and analyzed for the expression of viral capsid proteins using antibodies specific to each virus (Material and Methods).
(TIF)

**S8 Fig. Conservation of cleavage targets across enteroviruses.** The list of proteins targeted for cleavage in eGFP/CVB3-infected HeLa cells were checked for their presence in the proteomics dataset of the indicated viruses. The tiles, indicating the detection of the proteins shown on left, are color-coded for visual clarity.
(TIF)

**S9 Fig. Validation of proteins targeted by multiple enteroviruses.** HeLa cells infected with CVB3 (eGFP/CVB3), PV (poliovirus type 1), HRV (human rhinovirus A16), EV70 (enterovirus D-70), EV71 (enterovirus A-71), VEEV (Venezuelan equine encephalitis virus) or VSV (vesicular stomatitis virus) were lysed at the indicated times, and equal amounts of total protein were subjected to western blot with the indicated antibodies. The black solid and red dotted arrows indicate the full-length protein and the cleavage products, respectively. Some of the western blot images are shown in Fig 5.
(TIF)

**S10 Fig. Testing protein cleavages in multiple cell types.** HeLa (cervical epithelial) cells infected with eGFP/CVB3 and PV for 6h, Caco-2 (intestinal epithelial) cells infected with eGFP/CVB3 and PV for 8h, NPC (neural progenitor cells) infected with eGFP/CVB3 and PV for 8h, RD (rhabdomyosarcoma) cells and SK-N-SH (brain epithelial) cells infected with PV for 8h were analyzed for the cleavage of indicated proteins. The full-length protein is shown with black solid arrows, while the cleavage products with red dotted arrows.
(TIF)

**S11 Fig. LSM14A is cleaved in stem cell-derived neurons infected with enteroviruses.** Immunofluorescence images: embryonic stem cell-derived neurons were stained for TUJ1, a neuron-specific protein (left panel), and with DAPI as a nuclear stain (middle panel). The right panel shows the merged image. Western blot: The neurons were infected with eGFP/CVB3 or PV for 7h followed by western blot with anti-LSM14A antibody. The solid black arrow shows the full-length protein while the red dotted arrow indicates the C-terminal cleavage product.
(TIF)

**S12 Fig. MAVS activates ISRE in the absence of SeV infection.** 293T cells, seeded into 48-well plates at a density of 30,000 cells/well, were transfected with the ISRE-FLuc and RL-RLuc reporter plasmids along with GFP plasmid (vector) or increasing concentrations of MAVS plasmid (6.25 and 12.5 ng/well). Twenty-four hours later, the cells were either left uninfected or infected with SeV for 24h followed by the reporter assays as described in Fig 7A. V, vector.
(TIF)

**S13 Fig. Characterization of LSM14A-TEVP.** (A) 293T cells were transfected with the indicated concentrations of the vector, LSM14A-WT, or LSM14A-TEVP plasmids and infected with SeV. The reporter assay was performed as described in Fig 7A. (B) The 293T cells containing dox-inducible TEVP were transfected with the increasing amount of the V5-LSM14A--TEVP-HA plasmid and the TEVP expression was induced using the indicated concentrations of doxycycline. The LSM14A cleavage was monitored by western blot with anti-HA antibodies. The small amount of cleavage product seen in the untreated cells is reflective of the leakiness of the dox promoter.
(TIF)

**S1 File. Proteins cleaved in CVB3-infected HeLa cells at 4h post-infection.** The list of cleavage peptides and their corresponding proteins is shown. The proteins cleaved at 2A$^{pro}$ motifs are highlighted in blue while those cleaved at 3C$^{pro}$ motifs are highlighted in red. Further details can be found in the "Read me" sheet of the excel file.
(XLS)

**S2 File. The cleavage peptides and their corresponding proteins identified in HeLa cells infected with CVB3, PV, HRV, EV70, and EV71.** Instructions to read the data are provided

in the "Read me" sheet of the excel file.
(XLSX)

## Acknowledgments

Vincent Racaniello from Columbia University, New York, kindly provided enterovirus 70 cDNA clone pDNE9. Human rhinovirus A16 strain was a generous gift from Ann Palmenberg of the University of Wisconsin, Madison. A plasmid carrying subtiligase was kindly provided by James A. Wells of the University of California, San Francisco. We thank Veit Hornung of the Universitat Bonn, Germany, for providing 293T cells deficient for RIG-I, MDA5, MAVS, and STAT1. We are grateful to the Protein Science Facility at Karolinska Institutet/SciLifeLab for CVB3 protease purification. We thank Thomas B. Kepler and Axin Hua of the Boston University School of Medicine for assistance with bioinformatic analysis. We also thank Ellen Castillo and Arnella Webson for laboratory assistance, and William M. Schneider and Catherine Jones for critical reading of the manuscript.

## Author Contributions

**Conceptualization:** Mohsan Saeed, Margaret R. MacDonald, Charles M. Rice.

**Data curation:** Mohsan Saeed, Henrik Molina.

**Formal analysis:** Mohsan Saeed, Milica Tesic Mark, Maxwell L. Neal, John D. Aitchison, Henrik Molina.

**Funding acquisition:** Mohsan Saeed, Malin Flodström-Tullberg, Charles M. Rice.

**Investigation:** Mohsan Saeed, Nicholas T. Hertz, Xianfang Wu, Alison W. Ashbrook.

**Methodology:** Mohsan Saeed, Nicholas T. Hertz, Kierstin Bell, Henry A. Zebroski.

**Project administration:** Mohsan Saeed, Charles M. Rice.

**Resources:** Malin Flodström-Tullberg, Charles M. Rice.

**Software:** Maxwell L. Neal, John D. Aitchison, Henrik Molina.

**Supervision:** Charles M. Rice.

**Validation:** Mohsan Saeed, Sebastian Kapell.

**Visualization:** Mohsan Saeed, Maxwell L. Neal, John D. Aitchison.

**Writing – original draft:** Mohsan Saeed.

**Writing – review & editing:** Xianfang Wu, Alison W. Ashbrook, Henrik Molina, Charles M. Rice.

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
