## [Decision Letter · Decision Letter 0]

15 Jun 2020

Dear Charlie,

Thank you very much for submitting your manuscript "Defining the proteolytic landscape during enterovirus infection" for consideration at PLoS Pathogens. As with all papers reviewed by the journal, your manuscript was reviewed by members of the editorial board and by three independent reviewers. In light of the mostly-positive reviews (below this email), we would like to invite the re-submission of a revised version that takes into account the reviewers' comments. Beyond the editorial suggestions/queries of all three reviewers, Reviewer #1 had several substantive criticisms that will need to be addressed with additional experiments.

We cannot make any decision about publication until we have seen the revised manuscript and your response to the reviewers' comments, including the requested experimentation. Your revised manuscript may be sent to reviewers for further evaluation.

Please prepare and submit your revised manuscript within 60 days. If you anticipate any delay, please let us know the expected re-submission date by replying to this email. Please note that revised manuscripts received after the 60-day due date may require evaluation and peer review similar to newly submitted manuscripts.

Best regards,

Bert Semler

Guest Editor

PLOS Pathogens

Guangxiang Luo

Section Editor

PLOS Pathogens

Kasturi Haldar

Editor-in-Chief

PLOS Pathogens

orcid.org/0000-0001-5065-158X

Michael Malim

Editor-in-Chief

PLOS Pathogens

orcid.org/0000-0002-7699-2064

Reviewer's Responses to Questions

**Part I - Summary**

Reviewer #1: The manuscript by Saeed et al examines the host proteolytic network in enterovirus infected cells using an unbiased positive-selection N-terminomics approach. Specifically, this approach (developed by Jim Wells's group) involves appending a biotin-peptide to neo-N-terminal proteins using subtiligase. Although this method has been used successfully to identify cleaved proteins under stress conditions and in in vitro studies using a viral protease (flavivirus), this is the first study to apply this approach in virus-infected cells and (to my knowledge) is the first N-terminomics approach in virus infected cells. The study is comprehensive, applying this approach in five different enterovirus infections in a time dependent manner in order to capture temporal cleavage events. Through extensive (impressive) immunoblotting, they validate 33 novel targets in infected cells and in in vitro cleavage assays using cell lysates. Not surprisingly, they identify targets with a wide range of biological functions and reveal common targets between all enteroviruses and targets that are specific to a subset of enterovirus infections as well as those that are 3C and 2A protease specific. Overall, this part of the study is impressive, providing the first comprehensive view of the cleaved proteome in enterovirus infected cells with extensive validation and analysis of the cleavage sites. Further, this study shows the power and utility of N-terminomic approaches in revealing novel virus host interactions. Also a note, the authors optimized the subtiligase approach to use 10 fold less lysate (~3 mg), which was a barrier in previous protocols.

In order to show the functional significance of a 2Aprotease-cleaved target, they chose to follow up with LSM14A, a P bodies factor that has been implicated in innate immune signaling in SeV and HSV infection. Using overexpression studies and KO cell lines, the authors show that LSM14A expression activates ISRE transcription under SeV infection (not in mock cells) and that the cleaved N or C term fragments do no induce. Using a clever approach, they replace the 2A cleavage site within a tagged LSM14A with a TEV cleavage site and show that induced expression of TEV protease in cells led to loss of ISRE activation. Although these results implicate LSM14A in innate immune signaling, unfortunately, the exact role of LSM14A in this signaling pathway and how cleavage would affect this is not clear. Finally, the importance of cleavage of LSM14A in enterovirus infected cells could not be shown.

In sum, this a comprehensive study that identifies novel cleaved proteins in infected cells using an innovative N-terminomics approach. The identification of these cleaved proteins should will undoubtedly be seminal for future studies in understanding virus host interactions in enterovirus infections. However, there are several outstanding issues with LSM14A that need to be addressed.

Reviewer #2: In this manuscript, Saeed et al. identify sites of host protein cleavage upon enterovirus infection, validate a number of these sites orthogonally, and examine one proteolytic substrate in more detail. It is a substantive and thorough piece of work and perhaps long overdue for the field. It is worthy of publication. None of the following comments detract measurably from this conclusion.

Reviewer #3: Review Plos Path C. Rice EV Proteases

This manuscript describes the first attempted comprehensive examination of the extent and breadth of human proteins cleaved during infection by the two enterovirus proteinases. The authors employed technical strategies based on a bioengineered protein ligase, subtiligase, to label and capture nascent C-terminal protein cleavage fragments which were identified by LC-MS/MS.

This work indirectly builds on prior findings by others using similar strategies to ID cleavage products of poliovirus and CVB3 3Cpro generated in vitro. This manuscript goes further by using similar technical approaches to catalogue cleavages by both virus proteases in virus infections with five different enteroviruses (PV, EVD70, EVA71, HRVA16, CVB3). Several identified protease cleavages were validated by expression of proteases in cells and immunoblot approaches to examine protein cleavage. Other validation included in vitro cleavage with purified proteinases to confirm 38 targets. The authors also compared cleavage products generated in Caco2, RD, SK-N-SH cells with Hela cell cleavage products, which confirmed similar cleavage across cell types. Other findings presented are the exact cleavage sites targeted by each virus proteinase, which mostly confirmed the previous known cleavage site specificity of each proteinase but revealed greater substrate recognition variability, particularly with 2Apro. These exceptions to known cleavage site requirements were noted and are not unexpected. There is no mention of previously documented conformational constraints that also govern substrate specificity.

The authors focused more on one newly described cleavage target LSM14a, a factor in P-bodies with a role in innate immunity. LSM14a was found to be cleaved by 2Apro, a series of experiments indicated some functional impact of this cleavage in disrupting innate immune activation. This included engineering an TEV protease cleavage site where 2Apro cleaves to demonstrate that cleavage at the same site in LSM14a with a different protease disrupted its function in transcriptional activation of ISRE. The authors then further pursue the mechanisms of LSM14a activation of innate responses, performing experiments with knockdown of MAVS, RIG-I, STAT1. These experiments, clarified, but did not reveal the true function of LSM14a, indicating LSM14a is not essential for RIG-I activation but may serve as a co-factor.

Overall the manuscript describes a large amount of work and resulting data that provides a useful catalog of viral proteinase targets, and shows most are conserved between viruses but not all. Ultimately, many of the new targets, perhaps most, will represent noise, by-stander targets that may have no real impact on virus replication advantages. This is revealed in the breadth of cell pathways of the proteins identified. However some new protease targets could reveal new mechanisms of import in virus pathogenesis. However, the authors note that in toto, the net results from multiple virus cleavage targets further reduce host gene expression beyond the well known virus shutdown of translation and transcription. The repository of information here may be of general interest to virologists in other systems. Other than the increase in the catalog of proteins known to be targeted by enteroviruses, there are few findings concerning enterovirus pathogenesis beyond the LSM14a results presented.

**Part II – Major Issues: Key Experiments Required for Acceptance**

Reviewer #1: My comments are more to improve the manuscript:

1) The functional significance of LSM14A cleavage was examined in SeV infection and following ISRE activation. In Fig 7A, it is not shown that LSM14A containing the TEV protease site is cleaved by TEV. It is important to show that LSM14A is cleaved and the protein products are similar to that in enterovirus infected cells.

2) In Fig. 6A/B, mutations of the P1' position on LSM14A does not prevent cleavage and in fact, leads to a larger cleaved C-terminal fragment protein. Is there another cryptic 2A site nearby? It is important to (i) show that the mutation does block 2A cleavage in an in vitro cleavage assay and (ii) to try and track down this other cleavage site. If you look at the immunoblots of LSM14A in Figures 2A and 3A, there is clearly another slower migrating band, suggesting another cleavage site. Even in It is important as all subsequent N-term and C-term LSM14A studies in Fig 7 are based on one of these cleavage sites. In Fig 7, is LSM14A cleaved under SeV infection and what is the status (immunoblotting) of expressing the N-term or C-term fragments in infected cells.

An interesting question is to monitor LSM14A localization (P bodies) by immunofluoresence. Perhaps LSM14A cleavage in infected cells disrupts its localization

3) The role of LSM14A is a bit confusing. Fig 7D, RIG-I is required for LSM14A-dependent ISRE activation, however, in Fig 7F, SeV can still induce ISRE independent of LSM14A.

Could expression of the N- or C-term fragment of LSM14A have a dominant negative effect on RIG-I signaling? in CVB3-infected cells? Does LSM14A interact with RIG-I or other signaling factors? Answering these questions may provide specific insights into the functional role for cleavage of LSM14A.

4) With the extensive analysis of the cleaved substrates, it would great to show the % of targets by viral protease (3C, 2A) vs other (cellular). How does this percentage change over the time course of infection?

5) P. 8, lines 88-90. This was a bit vague. It would be more informative to show the percent of proteins that showed the expected size. Although many targets were identified and validated to be cleaved under infection and in vitro, it is important to show that some of the cleavage sites identified are real and directly cleaved by 3C/2A. Mutation of the cleavage sites (within a tagged protein) should prevent cleavage in infected cells and in vitro. This was only attempted for LSM14A. Alternatively, it is more direct to use purified, recombinant protein (can be bought) in in vitro cleavage assays.

6) There were several statements that were too strong "enterovirus-mediated cleavage renders LSM14A inactive". " enterovirus 2Apro-mediated cleavage blunted the ability of this protein to contribute to antiviral immune signaling." p. 17 " However, when LSM14A is cleaved by 2Apro, it can no longer mediate immune activation, indicating a loss of function."

These are not true, there is no evidence of this in enterovirus infected cells or results that directly show that 2Apro cleavage of LSM14A inactivates immune activation.

Reviewer #2: None.

Reviewer #3: (No Response)

**Part III – Minor Issues: Editorial and Data Presentation Modifications**

Reviewer #1: Other comments:

Figure 1B - are the proteins grouped in a particular manner? More clarification is needed.

Figure 6A - it would be good to have a cartoon above with the P1-P4/P1'-P4' amino acids to help understand how the mutations would disrupt cleavage.

P. 5, line 103. The N-terminomics approach cited in ref 21 is not similar but is distinct using a negative-selection approach whereas the subtiligase is a positive-selection.

P. 8, line 61 - missing reference for cleavage site of caspases

Reviewer #2: The study raises interesting future questions about how 2A site specificity is achieved. It is also interesting to consider how infection may modulate the cell’s natural peptidomic landscape via, for example, the TAP transporters (the MS portion of the study does not seem to require neo-N-termini to represent a simple bisection, or for peptides to be intracellular).

The initial portion of the study, namely the N-terminomics of CVB-3-infected HeLa extract via peptide ligation to unblocked N-termini, was likely challenging to perfect despite optimization: Here, the presence of a substantial unmodified proteome, even after the enrichment steps, is suggested by the setting of Abu as a variable modification in order to achieve acceptable FDR. In future work, this could perhaps be addressed by implementing an uninfected/infected isotope split. In this regard (P.6/Line 19), it is not clear what is meant by “multiplexed samples” since it appears they were not multiplexed according to M&M.

The authors identify unique human targets from single peptides, and the latter would need to be of sufficient length and quality (of fragmentation spectra) to evince unique matches in the human proteome. In this regard, it is not clear whether a peptide length filter was imposed: Parental spectra scanned from a low m/z of 140 Th, sufficiently low to identify Abu-tagged peptides just 2 aa in length. The fragmentation method (presumably HCD, since it was a Q-E?) does not seem to be specified. The authors identify cleavage sites from the downstream portion of the cleavage site only. Albeit unlikely, is there an assumption data do not include secondary sites from exopeptidase nibbling?

In the following comments, note that the first digit of all line numbers was cut off on my pdf. Also, the supp data did not seem to be available for review (at least in the links/logins I could find). The authors fund these oversights by the journal.

P.8/Line x72: Were the known protein targets confirmed at the level of target site within the protein?

P.9 Line x80 vs. line x84: Was it 81 or 90 proteins?

P.19/Line x91: How do the authors suppose that enteroviral proteases target intercellular junctions?

In Fig. 2A, might virus-induced loss of function/gain of function correspond to quantitative cleavage/incomplete cleavage respectively? Is it clear why cleavages can be detected prior to the first appearance of eGFP?

P.28/line 601: Induction temperature seems missing.

P.29 line 623: 150 microG/microL -> 15 microG/microL

LC-MS/MS section of M&M should specify nanocolumn dimensions/packing/details, along with high m/z (scan range) and fragmentation method. “tandem MS experiments” -> “Tandem MS”.

The authors could have perhaps included a sentence, somewhere, on the known functions of p-bodies. (e.g. in cytoplasmic mRNA turnover).

Wording:

P.4/Line x78 (“2A is only responsible for its own cleavage from the upstream structural protein”) should perhaps be re-worded given 2A’s diverse set of cellular substrates.

P.7 Lines x34/x35: “Intensity of labeling” could perhaps be replaced with “Diversity of labeling”.

P.3/Line x66: Close bracket.

P.2/Line x44: Add “a”. P.9/Line 81: were -> are.

P.14/Line x99: “Showed” -> “indicated”

P.17/line x47 “viruses” -> “enteroviruses” (since VEEV & SeV seem not to).

P.17/line x51 “Novel proteins” -> “novel substrates”.

P.18/Line x70 “Two C-terminal domains” -> “Two domains C-terminal to the unstructured region”.

Reviewer #3: (No Response)

PLOS authors have the option to publish the peer review history of their article (what does this mean?). If published, this will include your full peer review and any attached files.

Reviewer #1: No

Reviewer #2: No

Reviewer #3: No
---

## [Decision Letter · Decision Letter 1]

24 Aug 2020

Dear Charlie,

We are pleased to inform you that your revised manuscript 'Defining the proteolytic landscape during enterovirus infection' has been provisionally accepted for publication in PLOS Pathogens.

Best regards,

Bert L. Semler

Guest Editor

PLOS Pathogens

Guangxiang Luo

Section Editor

PLOS Pathogens

Kasturi Haldar

Editor-in-Chief

PLOS Pathogens

orcid.org/0000-0001-5065-158X

Michael Malim

Editor-in-Chief

PLOS Pathogens

orcid.org/0000-0002-7699-2064

Reviewer Comments (if any, and for reference):

Reviewer's Responses to Questions

**Part I - Summary**

Reviewer #1: The author has responded to most of my comments satisfactorily and made editorial changes in the certain concluding statements to more reflect the results.

Reviewer #2: (No Response)

**Part II – Major Issues: Key Experiments Required for Acceptance**

Reviewer #1: (No Response)

Reviewer #2: (No Response)

**Part III – Minor Issues: Editorial and Data Presentation Modifications**

Reviewer #1: (No Response)

Reviewer #2: I have nothing really to add to my initial review. In one or two cases the authors did not fully comprehend my queries e.g. the precursor mass range question - precursor scan ranges are always fixed for all precursor spectra as opposed to the fragmentation spectra (the person revising the manuscript was presumably not the mass spec guy), but I think this is not worth any further pursuit, so I hit "accept".

PLOS authors have the option to publish the peer review history of their article (what does this mean?). If published, this will include your full peer review and any attached files.

Reviewer #1: No

Reviewer #2: No

---

## [Editor Report · Acceptance letter]

24 Sep 2020

Dear Dr. Rice,

We are delighted to inform you that your manuscript, "Defining the proteolytic landscape during enterovirus infection," has been formally accepted for publication in PLOS Pathogens.

Best regards,

Kasturi Haldar

Editor-in-Chief

PLOS Pathogens

orcid.org/0000-0001-5065-158X

Michael Malim

Editor-in-Chief

PLOS Pathogens

orcid.org/0000-0002-7699-2064